# Enabling Large Language Models to Generate Text with Citations

**Tianyu Gao   Howard Yen   Jiatong Yu   Danqi Chen**
Department of Computer Science & Princeton Language and Intelligence
Princeton University
{tianyug,hyen,jiatongy,danqic}@cs.princeton.edu

## Abstract

Large language models (LLMs) have emerged as a widely-used tool for information seeking, but their generated outputs are prone to hallucination. In this work, our aim is to allow LLMs to generate text with *citations*, improving their factual correctness and verifiability. Existing work mainly relies on commercial search engines and human evaluation, making it challenging to reproduce and compare different modeling approaches. We propose **ALCE**, the first benchmark for **A**utomatic **L**LMs' **C**itation **E**valuation. ALCE collects a diverse set of questions and retrieval corpora and requires building end-to-end systems to retrieve supporting evidence and generate answers with citations. We develop automatic metrics along three dimensions—fluency, correctness, and citation quality—and demonstrate their strong correlation with human judgements. Our experiments with state-of-the-art LLMs and novel prompting strategies show that current systems have considerable room for improvement—For example, on the ELI5 dataset, even the best models lack complete citation support 50% of the time. Our analyses further highlight promising future directions, including developing better retrievers, advancing long-context LLMs, and improving the ability to synthesize information from multiple sources.[1]

## 1 Introduction

Large language models (LLMs; Brown et al., 2020; OpenAI, 2023) have gained increasing popularity as a tool for information seeking. While they generate engaging and coherent responses, their outputs are prone to hallucination and often contain factually incorrect information (Ji et al., 2023). This makes it harder for users to trust and verify LLM-generated outputs without any supporting evidence.

In this work, we study a new generation paradigm for LLMs, in which we require LLMs

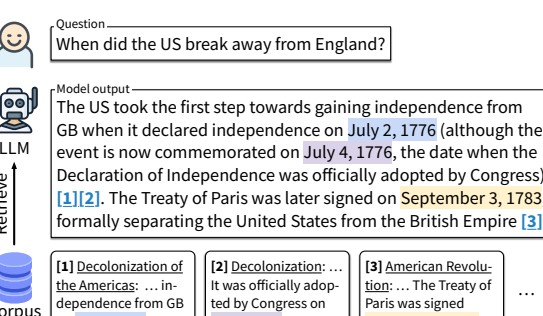

Figure 1: The task setup of ALCE. Given a question, the system generates text while providing *citing passages* from a large retrieval corpus. Each statement may contain multiple citations (e.g., [1][2]).

to provide *citations* to one or a few text passages for any statement they generate (Figure 1). Incorporating citations brings several benefits: (1) users can easily verify LLMs' claims with the provided citations; (2) LLMs can generate text that faithfully follows cited passages, which has the promise to improve correctness and alleviate hallucination.

Multiple commercial systems have adopted this paradigm: Bing Chat[2] and perplexity.ai[3] respond to user questions in natural language with references to Web pages. Nakano et al. (2021); Menick et al. (2022) share a similar motivation, but they mainly experiment with commercial search engines and closed-source models, making their results difficult to evaluate. Retrieval-augmented LMs (Borgeaud et al., 2022; Izacard et al., 2022) incorporate retrieved passages during both training and inference, but do not guarantee faithfulness to retrieved passages or explicitly provide citations. Additionally, previous studies mostly rely on human evaluation (Nakano et al., 2021; Menick et al., 2022; Liu et al., 2023), which is expensive and difficult to reproduce. We argue that the absence of automated evaluation hinders the advances of such systems.

---

[1] Our code and data are available at https://github.com/princeton-nlp/ALCE.

[2] https://www.bing.com/new

[3] https://www.perplexity.ai

| Dataset | Corpus (#passages) | Question type | Example |
|---------|--------------------|--------------------|---------|
| ASQA | Wikipedia (21M) | Factoid (ambiguous) | **Q**: When did the US break away from England?
**A**: The US declared independence on July 2, 1776 [1][2] ... The Treaty of Paris was later signed on September 3, 1783 [3]. |
| QAMPARI | Wikipedia (21M) | Factoid (list) | **Q**: Which films have Gong Li as a member of their cast?
**A**: The Story of Qiu Ju [1], Farewell My Concubine [2], The Monkey King 2 [3], Mulan [3], Saturday Fiction [3] ... |
| ELI5 | Sphere (899M) | Why/How/What | **Q**: How do student loans affect getting a mortgage?
**A**: Student loans can affect the debt to income ratio [1], which is a key factor in determining the amount that ... [2][3] |

Table 1: The three datasets used in our ALCE benchmark. These datasets cover a wide range of question types and the corresponding corpora span from Wikipedia to Web-scale document collection.

We present **ALCE**, the first reproducible benchmark for *automatically* evaluating LLMs' generations with citations. ALCE assumes a natural-language question and a retrieval corpus, and requires building end-to-end systems to retrieve relevant passages from the corpus, generate a response to the question, and cite corresponding supporting passages. We compile three datasets that cover different types of questions and corpora—ASQA (Stelmakh et al., 2022), QAMPARI (Rubin et al., 2022), and ELI5 (Fan et al., 2019)—as shown in Table 1. Different from previous benchmarks (Lee et al., 2019; Bohnet et al., 2022), ALCE evaluates long-text generation, focusing on automatically evaluating citation quality, and allows citing *multiple* passages for individual statements.

We design automatic evaluation methods in three dimensions: **fluency**, **correctness**, and **citation quality**. Specifically, we use MAUVE (Pillutla et al., 2021) to measure fluency, propose tailored correctness metrics for each dataset, and adopt a natural language inference (NLI) model (Honovich et al., 2022) to measure citation quality. We showcase how the three dimensions together contribute to a robust evaluation, preventing systems from exploiting shortcuts. Additionally, we conduct human evaluation and demonstrate a strong correlation with our automatic metrics.

We experiment on multiple systems with state-of-the-art LLMs and retrievers and also propose novel prompting strategies to synthesize retrieved text into text generation. Although all systems are capable of providing fluent and coherent responses, there remains substantial room for improvement in terms of correctness and citation quality: For example, on the ELI5 dataset, around 50% generations of our ChatGPT and GPT-4 baselines are not fully supported by the cited passages. Additionally, we find that (1) a closed-book model (generating answers without accessing any retrieved documents)

with post-hoc citing achieves good correctness but much worse citation quality; (2) although interactive retrieval approaches (Yao et al., 2023; Schick et al., 2023) offer more flexibility in when/what to retrieve, they do not improve the performance on this challenging benchmark; (3) summarizing the retrieved passages in a shorter text improves correctness but not citation quality; (4) reranking multiple generations boosts citation quality measured by human evaluation; (5) incorporating more retrieved passages in context does not help Chat-GPT but improves GPT-4 performance.

Our extensive analyses highlight three major challenges of building LLMs to generate text with citations: (1) the retrieval quality is crucial to the final performance and has substantial room for improvement; (2) LLMs' limited context window restricts the number of passages they can incorporate; (3) current LLMs struggle to synthesize multiple documents in context without being distracted by irrelevant ones, although better instruction tuning brings significant improvement. These challenges pose promising research directions for developing better systems integrating retrieval and LLMs.

## 2 Task Setup and Datasets

Our task is formalized as follows: Given a query $q$ and a corpus of text passages $\mathcal{D}$, the system is required to return an output $\mathcal{S}$, which consists of $n$ statements $s_1, ..., s_n$, and each statement $s_i$ cites a list of passages $\mathcal{C}_i = \{c_{i,1}, c_{i,2}, ...\}$[4], where $c_{i,j} \in \mathcal{D}$. In this work, we segment LLMs' output into statements by sentence boundaries.[5] While LLMs may include sentences that do not require a citation, such as "*I'm happy to help*", we observe that almost all sentences that LLMs output provide

---

[4]In practice, we allow at most 3 citations for each statement as more citations usually do not help.

[5]QAMPARI requires a list as the answer, and we choose each entity in the generated list as a statement.

valuable information and require citations, similar to findings in Liu et al. (2023). In this work, citations are enclosed by box brackets such as [1][2].

We divide the corpus $\mathcal{D}$ into 100-word passages following previous works on open-domain question answering (Karpukhin et al., 2020; Petroni et al., 2021; Piktus et al., 2021), in contrast to commercial systems like Bing Chat, which cite entire Web pages. We take 100-word passages because it is easier for humans to verify, and allows for more retrieved passages to fit in LLMs' limited context.

We choose QA datasets so that (1) they contain factual questions, in which references are important; (2) questions require long-text answers that cover multiple aspects; (3) answering the questions requires synthesizing multiple sources. We select three datasets (Table 1) and introduce them below. See §B for additional statistics.

**ASQA** (Stelmakh et al., 2022) is a long-form factoid dataset. As shown in Figure 1, each question is an ambiguous question from AmbigQA (Min et al., 2020) that requires multiple short answers to cover different aspects, and the dataset provides a long-form answer that covers all short answers. Since most questions can be answered by Wikipedia, we use the 2018-12-20 Wikipedia snapshot as $\mathcal{D}$.

**QAMPARI** (Rubin et al., 2022) is a factoid QA dataset constructed from Wikipedia, where the answer is a list of entities that are drawn from different passages. Same as ASQA, we use the 2018-12-20 Wikipedia as the corpus.

**ELI5** (Fan et al., 2019) is a long-form QA dataset built on the Reddit forum "Explain Like I'm Five".[6] Most ELI5 questions are how/why/what questions that require long answers and multiple passages as evidence. Due to the diverse topics discussed in the questions, we use Sphere (Piktus et al., 2021)—a filtered version of Common Crawl[7]—as the corpus. The ELI5 dataset is widely used in related work due to its challenging nature (Nakano et al., 2021; Menick et al., 2022; Liu et al., 2023).

We randomly select 1,000 examples from the development set of each dataset for ALCE. Our benchmark primarily assesses the citation capabilities of existing LLMs and does not provide training data, as there are no available examples that provide supervision for citations in these datasets.

---

[6]https://www.reddit.com/r/explainlikeimfive/
[7]https://commoncrawl.org. We also filter out any Web pages from Reddit.

## 3 Automatic Evaluation

Our benchmark measures the following three dimensions of system responses:

- **Fluency**: whether the model's generated text is fluent and coherent.
- **Correctness**: whether the answer is accurate and covers all aspects of interest.
- **Citation quality**: whether the answer is well supported by the cited passages and no irrelevant passages are cited.

In the following, we present automatic metrics for each dimension and discuss why the combination of the three metrics provides a robust evaluation.

### 3.1 Fluency

We use MAUVE (Pillutla et al., 2021) to evaluate the fluency of the output (§C). We deploy MAUVE for ASQA and ELI5 and omit it for QAMPARI, as QAMPARI only requires a list of short answers as the response and LLMs consistently adhere to the format in our experiments. As MAUVE is sensitive to output length and text style, and most LLMs are capable of producing fluent text, we mainly employ it as a sanity check as long as the MAUVE scores are high enough.

### 3.2 Correctness

Our objective is to measure the informativeness and utility of the generation to the question. Liu et al. (2023) propose to directly evaluate *perceived utility* by humans, a process difficult to automate. Therefore, we use correctness—whether the response is accurate compared to a ground truth answer—as a proxy. Evaluating the correctness of long-form generation is a challenging task (Krishna et al., 2021), and we describe our strategy for each dataset below. Figure 2 illustrates the metrics and we include additional implementation details in §C.

For **ASQA**, we follow Stelmakh et al. (2022) and calculate the recall of correct short answers by checking whether the short answers (provided by the dataset) are exact substrings of the generation (*exact match recall*; EM recall).

For **QAMPARI**, we follow Rubin et al. (2022) and calculate the *precision* and *recall* of the model prediction, by checking the exact match to the gold answer list. We add one additional adjustment: considering that users often want to know only a few example answers of the question, our evaluation considers recall to be 100% if the prediction includes at least 5 correct answers (*recall-5*).

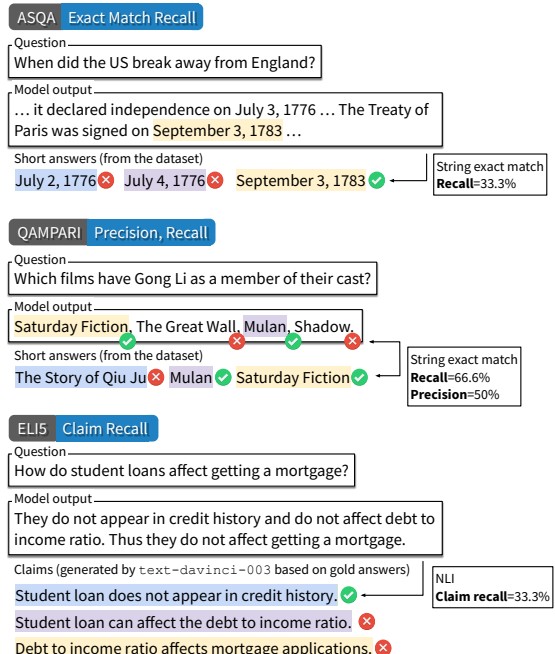

Figure 2: Evaluation of correctness (details in §3.2).

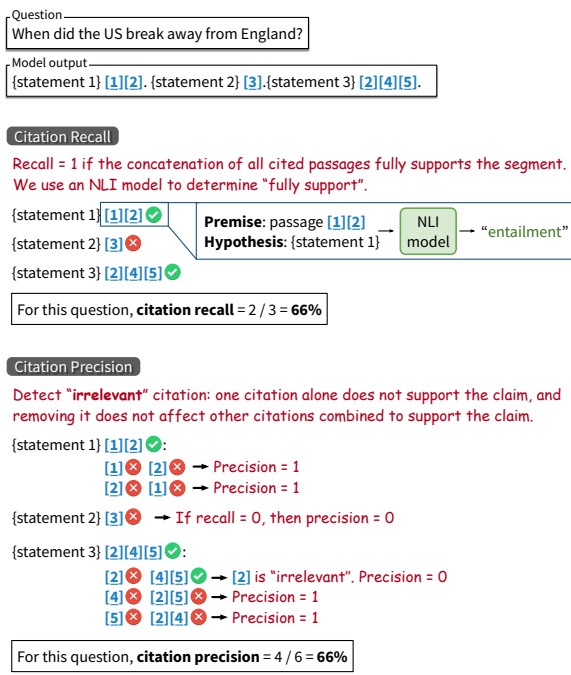

Figure 3: Evaluation of citation quality (details in §3.3). We use an NLI model to verify whether a statement is supported by its citations.

Unlike ASQA and QAMPARI, the **ELI5** dataset does not provide short entity answers. Fan et al. (2019) use ROUGE for evaluation, which does not reflect the correctness well (Krishna et al., 2021; §A). Inspired by works in summarization evaluation (Zhang and Bansal, 2021; Kamoi et al., 2023; Wang et al., 2020), we use Instruct-GPT (text-davinci-003; Ouyang et al., 2022) to generate three "sub-claims". Then we use TRUE[8] (Honovich et al., 2022), a T5-11B (Raffel et al., 2020) model fine-tuned on a collection of natural language inference (NLI) datasets, to check whether the model output entails the sub-claims (*claim recall*). TRUE targets factual correctness and has been used by previous works in similar context (Bohnet et al., 2022; Gao et al., 2023). We demonstrate that claim recall provides a more accurate measure of correctness than existing metrics (more details in §A).

### 3.3 Citation Quality

We evaluate citation qualities using two metrics: (1) *citation recall*, which determines if the output is entirely supported by cited passages, and (2) *citation precision*, which identifies any irrelevant citations. Although we prioritize citation recall as it entails a well-supported and truthful answer, enhancing precision is crucial for better user satisfaction, reducing the need for human review of extraneous

passages. Figure 3 provides an illustrated example.

We use the NLI model TRUE (Honovich et al., 2022) again to automatically examine whether the cited passages entail the model generation. We conduct human evaluation (§6) to demonstrate strong human correlation of our metric.

**Citation recall.** We calculate the citation recall of *each statement* (0 or 1) and average over all statements in the model response. For each statement $s_i$, its citation recall is 1 if and only if there is at least one citation ($\mathcal{C}_i \neq \emptyset$) and $\phi(\text{concat}(\mathcal{C}_i), s_i) = 1$, where $\phi(\text{premise}, \text{hypothesis})$ is the NLI model that outputs 1 if the premise entails the hypothesis, and 0 otherwise; $\text{concat}(\mathcal{C}_i)$ concatenates all passages in $\mathcal{C}_i$ together (details in §C). The NLI evaluation is in accordance with the *attributable to identified sources* (AIS) framework (Rashkin et al., 2023): $\phi(\text{concat}(\mathcal{C}_i), s_i) = 1$ implies that $s_i$ is true based solely on $\text{concat}(\mathcal{C}_i)$.

**Citation precision.** Our citation precision evaluation detects citations that are irrelevant, but it does not require citing a minimal set. We follow this design because human writing often cites redundant sources to enhance credibility; human readers may also appreciate multiple citations, especially when it pertains to critical claims such as medical advice.

We calculate the citation precision for *each citation* (0 or 1) and average over all citations in the

---

[8] https://huggingface.co/google/t5_xxl_true_nli_mixture. Details in §C.

```
Instruction: Write an accurate, engaging, and
concise answer for ...

<Retrieve for the question>
Document [1](Title: American Decolonization)
...
Document [2](Title: Decolonization) ...
Document [3](Title: American Revolution) ...
...

Question: When did US break away from England?
Answer: The United States took the first step
towards gaining independence ... [1][2]. The
Treaty of Paris was later signed ... [3].
```

Table 2: An example of our VANILLA method. Different colors represent prompt, model generation, and <actions>. We also provide two in-context demonstrations before the test example.

response. We first define if a citation is "irrelevant". Intuitively, a citation $c_{i,j}$ is "irrelevant" if (a) $c_{i,j}$ itself cannot support $s_i$ and (b) removing $c_{i,j}$ does not affect the rest of the citations to support $s_i$. Formally, $c_{i,j}$ is "irrelevant" if and only if

(a) $\phi(c_{i,j}, s_i) = 0$, AND

(b) $\phi(\text{concat}(\mathcal{C}_i \setminus \{c_{i,j}\}), s_i) = 1$.

$c_{i,j}$ has a precision of 1 if $s_i$ has recall=1 and $c_{i,j}$ is not irrelevant. For example (Figure 3), when $s_3$ cites three references [2][4][5] and recall=1, [2] is "irrelevant" if $\phi([2], s_3) = 0$ and $\phi([4][5], s_3) = 1$. For condition (b) to work, we set recall=1 as a prerequisite for precision= 1. Note that this algorithm overlooks the scenario when one citation partially supports the statement. We discuss the details in §E.

### 3.4 ALCE is Robust to Shortcut Cases

We showcase how the ALCE evaluation is robust to two possible shortcuts in §D: (1) using the top-1 retrieved passage as the response and citing itself, and (2) using the first two sentences of the top-1 passage. Both cases have almost-perfect citation scores, but (1) has low fluency due to its unnaturally long length compared to human answers, and (2) has low correctness due to low coverage.

## 4 Modeling

In this section, we discuss three major modeling components for an ALCE system—retrieval, synthesis, and post-editing.

### 4.1 Retrieval

We explore simple, off-the-shelf retrievers. We use dense retrievers for Wikipedia, including GTR (Ni

et al., 2022) and DPR (Karpukhin et al., 2020); we use BM25 for Sphere. For each question, we retrieve the top-100 passages.

### 4.2 Synthesis

We focus on how to prompt an LLM to interact with the retriever, and synthesize and cite the evidence (without fine-tuning internal parameters). One noteworthy challenge is that existing LLMs all have limited context window and thus can only fit a handful of passages.

**VANILLA.** We simply provide the model with the top-$k$[9] passages and instruct the model to cite accordingly (Table 2). We also use in-context learning (Brown et al., 2020) and prepend two demonstrations. The complete instruction is in Table 23.

**SUMM/SNIPPET.** With a 4K context window, we can at most safely fit $k = 5$ passages. As shown in Figure 4, top-5 retrieved passages can only cover 56.8% percent of the answers in ASQA.

To tackle this limitation, we propose to provide *summaries* or *snippets* of passages instead of the full text (summaries are abstractive but snippets are spans from passages). We acquire summaries and snippets by prompting ChatGPT with instructions (prompts in Table 25 and 26).[10] Then we replace all passages with summaries/snippets. Summaries or snippets significantly reduce the passage length, allowing for more passages to fit in: for ASQA, they reduce passage length by $6\times$ on average.

Though SUMM/SNIPPET allows for more retrieved passages, they are lossy compressions. To alleviate this problem, we propose INTERACT, an interactive prompting scheme to allow the model to check the full text of certain passages. At each step, the model can execute one of three actions: (1) "Check: Document [1][2]" to check the full text of the corresponding documents; (2) "Output:" to output a statement of the answer; (3) "End." to end the generation. §C provides more details.

**INLINESEARCH.** The above methods all display retrieval results at the beginning. In INLINESEARCH, we allow LLMs to call "search" during the generation process (Yao et al., 2023; Press et al., 2022; Jiang et al., 2023). At each step, the model can execute one of three actions: "Search:

---

[9]We can fit at most $k = 3$ for models with 2K window and at most $k = 5$ for models with 4K context window.

[10]We also query ChatGPT whether the passage is relevant to the question, and filter out passages that are "irrelevant".

```
Instruction: ...

<Retrieve for question "...">
Question: When did US break away from England?
Search: Declaration of Independence
<Search the query among the top-100 passages>
Document [1](Title: ...) ...
Output: The United States ... [1].
<Remove Document [1] from context>
Search: Treaty of Paris
<Search the query among the top-100 passages>
Document [3](Title: ...) ...
Output: The Treaty of Paris ... [3].
<Remove Document [3] from context>
End.
```

Table 3: An example of INLINESEARCH.

{query}" to search among the top-100 passages[11] by using GTR; the "Output" and "End" actions are the same as INTERACT. For each "Search" action, we display the best retrieved passage in the context. The passage is removed after one action to save context space. Table 3 shows an example.

**CLOSEDBOOK.** We also add a simple closed-book baseline, where the model is only prompted with the instruction and the question, without any retrieved passages provided. Consequently, this variant does not cite any evidences.

### 4.3 Post-editing

In this section we discuss two strategies for refining the output to further improve its quality.

**RERANK.** We randomly sample $n_{sample} = 4$ responses for each question, and select the best response using the automatic *citation recall* score. we expect RERANK to improve the citation quality.

**POSTCITE.** For each statement, we find the best matching passage among the top-100 retrieved passages using GTR and cite it. We combine this with CLOSEDBOOK in our experiments.

## 5 Experiments

We describe experiment details in §C. We use Chat-GPT (gpt-3.5-turbo-0301) with a 4K context window for most main experiments and ablations. We also report results with ChatGPT-16K (gpt-3.5-turbo-16k-0613) and GPT-4 (gpt-4-0613; 8K context window). For open-source models, we test LLaMA (Touvron et al., 2023a) and its instruction-tuned versions, including Alpaca (Taori et al., 2023), Vicuna (Chiang et al., 2023), and

| | Fluency | Correct. | Citation | |
|---|---|---|---|---|
| | (MAUVE) | (EM Rec.) | Rec. | Prec. |
| **ChatGPT** | | | | |
| VANILLA (5-psg) | 66.6 | 40.4 | 73.6 | 72.5 |
| w/ RERANK | 77.0 | 40.2 | **84.8** | **81.6** |
| SUMM (10-psg) | 70.0 | **43.3** | 68.9 | 61.8 |
| w/ INTERACT | 69.0 | 39.1 | 73.4 | 66.5 |
| SNIPPET (10-psg) | 69.8 | 41.4 | 65.3 | 57.4 |
| INLINESEARCH | 58.7 | 32.4 | 58.3 | 58.2 |
| CLOSEDBOOK | 52.7 | 38.3 | 26.7 | 26.7 |
| **GPT-4** (VANILLA prompting) | | | | |
| GPT-4 (5-psg) | 67.1 | 41.3 | 68.5 | 75.6 |
| GPT-4 (20-psg) | 64.9 | 44.4 | 73.0 | 76.5 |
| **LLaMA** (VANILLA prompting) | | | | |
| LLaMA-13B (3-psg) | 68.4 | 26.9 | 10.6 | 15.4 |
| Vicuna-13B (3-psg) | 82.6 | 31.9 | 51.1 | 50.1 |
| Chat-13B (5-psg) | 72.4 | 35.2 | 38.4 | 39.4 |
| Chat-70B (5-psg) | 88.3 | 41.5 | 62.9 | 61.3 |

Table 4: Experiments on ASQA. For CLOSEDBOOK, we use POSTCITE to get citations. $k$-psg: putting top-$k$ passages from the retrieval results into the context. Chat-13B and Chat-70B refer to LLaMA-2-Chat.

Oasst (Köpf et al., 2023). They all have a 2K context window. We use short instructions for LLaMA (Table 24) to save context budget. Additionally, we test LLaMA-2-Chat, which were also trained to follow instructions (Touvron et al., 2023b). These models have a context window of 4K tokens, which allows for 5 passages per question.

### 5.1 Main Results

We present the main results on three datasets in Table 4, 5, and 6 respectively (full results in §G.6). We first note that all models achieve good fluency scores (except some models on ELI5 mainly due to their longer generations). We summarize the main takeaways from the experiments below.

**VANILLA achieves strong performance.** Despite its simplicity, VANILLA (putting retrieved passages in context) achieves close-to-the-best performance among all prompting strategies.

**Using summaries or snippets improves correctness.** We see a universal trend that SUMM or SNIPPET improves correctness, though on ASQA and ELI5, such an improvement comes at a cost of citation quality due to the lossy compression. Combining INTERACT with SUMM/SNIPPET does not bring improvement, and we hypothesize that checking the full passages offers limited benefit and current LLMs are not proficient in an interactive usage.

**Retrieving text on the fly does not improve performance.** All datasets show that VANILLA outperforms INLINESEARCH on citation quality (and

---

[11]We do not search over the entire corpus because {query} may leave out certain context in the question and searching among the already-retrieved passages gives better results.

| | Correctness | | Citation | |
|---|---|---|---|---|
| | Rec.-5 | Prec. | Rec. | Prec. |
| **ChatGPT** | | | | |
| VANILLA (5-psg) | 20.8 | 20.8 | 20.5 | 20.9 |
| w/ RERANK | 22.8 | 21.4 | 21.2 | 21.4 |
| SUMM (10-psg) | 23.6 | 21.2 | **23.6** | **25.7** |
| SNIPPET (10-psg) | 24.5 | 21.5 | 22.9 | 24.9 |
| w/ INTERACT | 21.9 | **23.0** | 21.9 | 23.4 |
| INLINESEARCH | 17.2 | 20.4 | 14.9 | 14.9 |
| CLOSEDBOOK | **32.9** | 19.8 | 10.0 | 10.0 |
| **GPT-4** (VANILLA prompting) | | | | |
| GPT-4 (5-psg) | 22.2 | 25.0 | 25.9 | 27.0 |
| GPT-4 (20-psg) | 29.6 | 26.2 | 27.4 | 28.5 |
| **LLaMA** (VANILLA prompting) | | | | |
| LLaMA-13B (3-psg) | 9.7 | 9.1 | 6.7 | 7.1 |
| Vicuna-13B (5-psg) | 14.0 | 15.9 | 12.5 | 13.4 |
| Chat-13B (5-psg) | 21.1 | 18.2 | 9.6 | 9.7 |
| Chat-70B (5-psg) | 21.8 | 18.4 | 15.1 | 15.6 |

Table 5: Experiments on QAMPARI. "Rec.-5": we set the recall to be 100% if the prediction includes at least 5 correct answers.

| | Fluency | Correct. | Citation | |
|---|---|---|---|---|
| | (MAUVE) | (Claim) | Rec. | Prec. |
| **ChatGPT** | | | | |
| VANILLA (5-psg) | 57.2 | 12.0 | 51.1 | 50.0 |
| w/ RERANK | 56.1 | 11.4 | **69.3** | **67.8** |
| SUMM (10-psg) | 40.3 | 12.5 | 51.5 | 48.2 |
| SNIPPET (10-psg) | 62.9 | 14.3 | 50.4 | 45.0 |
| w/ INTERACT | 68.0 | 13.3 | 47.8 | 45.0 |
| INLINESEARCH | 49.7 | 13.4 | 45.6 | 43.7 |
| CLOSEDBOOK | 32.6 | **18.6** | 15.5 | 15.5 |
| **GPT-4** (VANILLA prompting) | | | | |
| GPT-4 (5-psg) | 38.4 | 14.2 | 44.0 | 50.1 |
| GPT-4 (20-psg) | 41.5 | 18.3 | 48.5 | 53.4 |
| **LLaMA** (VANILLA prompting) | | | | |
| LLaMA-13B (3-psg) | 50.0 | 3.9 | 3.1 | 5.3 |
| Vicuna-13B (3-psg) | 58.2 | 10.0 | 15.6 | 19.6 |
| Chat-13B (5-psg) | 34.7 | 13.4 | 17.3 | 15.8 |
| Chat-70B (5-psg) | 38.6 | 12.8 | 38.3 | 37.9 |

Table 6: Experiments on ELI5. We use *claim recall* for the correctness evaluation. Chat-13B and Chat-70B refer to LLaMA-2-Chat.

on correctness for ASQA and ELI5). By manually examining the examples, we find that it is challenging to ask detailed questions without seeing any passages. To improve INLINESEARCH, one may need to provide more context about the questions in advance or encourage the model to call retrievers with more detailed and diverse queries.

**RERANK boosts citation quality.** We observe that RERANK leads to consistent improvement in citation quality (on ASQA and ELI5). As the automatic scores may be biased in RERANK, we also conduct human evaluation (§6) and verify its effectiveness.

**CLOSEDBOOK+POSTCITE delivers strong correctness but poor citation quality.** CLOSEDBOOK outperforms VANILLA in correctness on ELI5 and QAMPARI, and has only a 2% gap on ASQA. However, CLOSEDBOOK cannot provide any citation; when combined with POSTCITE, the citation quality remains inadequate. For instance, citation recall of CLOSEDBOOK+POSTCITE is lower than VANILLA by 47% on ASQA.

To understand why CLOSEDBOOK achieves better correctness and why POSTCITE cannot deliver satisfying citation quality, we manually examine model outputs and find that: (1) open-book models are easily distracted by irrelevant passages and generate responses with lower correctness, a phenomenon also observed by Shi et al. (2023); (2) CLOSEDBOOK often generates texts that are correct but not similar to any retrieved passages, making it difficult to match a citation post-hoc.

**GPT-4 brings limited improvement but is better at using long context.** We evaluate GPT-4 with VANILLA and different numbers of passages (more results in §G.6). GPT-4 brings consistent (but limited) improvement on correctness, but often at a cost of citation quality. GPT-4 can also incorporate more passages due to its longer context window, which boosts both correctness and citation quality. On the contrary, including more passages with ChatGPT-16K does not improve the results (Table 7), suggesting that processing more passages is non-trivial and GPT-4 is better at synthesizing information from its long context than ChatGPT.

## 5.2 Comparison of Different LLMs

Table 7 compares different LLMs on ASQA using VANILLA (more results in §G.6). Notably, instruction-tuned models (Vicuna-13B and LLaMA-2-Chat) outperform the original LLaMA models in correctness and considerably enhance the citation quality. We observe that while the original LLaMA models are able to copy facts from the context, they struggle with accurately citing the sources or simply do not cite. Notably, the best open-source model, LLaMA-2-70B-Chat, achieves comparable correctness score as the OpenAI models, but still lags behind in citation quality.

## 5.3 Retrieval Analysis

The retrieval results play a crucial role to the correctness and the citation quality. Figure 4 presents the retrieval recall@$k$ with different datasets and

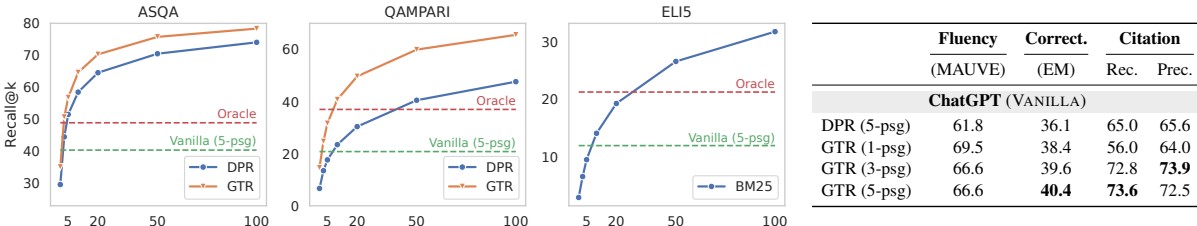

Figure 4: Retrieval recall@$k$ on ASQA (*EM recall*), QAMPARI (*recall-5*), and ELI5 (*claim recall*). Retrieval recall serves as an upper bound for model performance, and we compare them with two models' correctness results in the figure (dashed lines): "Vanilla (5-psg)" is ChatGPT VANILLA with top-5 passages in context; "Oracle" is the same model except that it uses 5 gold passages (§G.1), whose recall matches Recall@100 on all three datasets.

|  | Fluency | Correct. | Citation | |
|---|---|---|---|---|
|  | (MAUVE) | (EM Rec.) | Rec. | Prec. |
| **Open-source** (max #tokens=2K-4K) | | | | |
| LLaMA-13B (3-psg) | 68.4 | 26.9 | 10.6 | 15.4 |
| Vicuna-13B (3-psg) | 82.6 | 31.9 | 51.1 | 50.1 |
| Chat-13B (5-psg) | 72.4 | 35.2 | 38.4 | 39.4 |
| Chat-70B (5-psg) | 88.3 | 41.5 | 62.9 | 61.3 |
| **ChatGPT** (max #tokens=4K) | | | | |
| ChatGPT (3-psg) | 66.6 | 39.6 | 72.8 | 73.9 |
| ChatGPT (5-psg) | 66.6 | 40.4 | 73.6 | 72.5 |
| **ChatGPT-16K** (max #tokens=16K) | | | | |
| ChatGPT (5-psg) | 60.3 | 36.1 | 76.2 | 76.5 |
| ChatGPT (10-psg) | 56.3 | 36.7 | 75.3 | 75.0 |
| ChatGPT (20-psg) | 56.7 | 36.1 | 73.7 | 73.5 |
| **GPT-4** (max #tokens=8K) | | | | |
| GPT-4 (5-psg) | 67.1 | 41.3 | 68.5 | 75.6 |
| GPT-4 (10-psg) | 71.5 | 43.1 | 72.0 | 75.5 |
| GPT-4 (20-psg) | 64.9 | 44.4 | 73.0 | 76.5 |

Table 7: Comparison of different LLMs on ASQA (GTR+VANILLA). LLaMA-13B and Vicuna-13B have a context limit of 2,048 tokens, and thus can only use a short version of instructions and at most top-3 passages. Chat-13B and Chat-70B refer to LLaMA-2-Chat.

retrievers. As the number of passages increases, retrieval recall steadily improves. Additionally, Figure 4 shows the correctness performance of two models: (1) ChatGPT VANILLA with top-5 passages (our primary baseline); (2) an oracle version of the same model employing 5 gold passages (§G.1; the 5 gold passages match the retrieval recall@100). Notably, both models' correctness lags behind the corresponding retrieval recall (except for ELI5 top-5). The discrepancy suggests that despite the presence of accurate answers in context, LLMs struggle to utilize them in their outputs.

We compare the impact of different retrievers and different numbers of passages to LLMs. Figure 4 (right) shows that GTR outperforms DPR in both correctness and citation quality, emphasizing the importance of deploying better retrievers. Contrary to the retrieval recall trend in Figure 4, more passages in context do not yield substantial

improvement for ChatGPT. Specifically, correctness plateaus at top-1 passage and citation quality plateaus at top-3. GPT-4 (Table 7) exhibits an increasing trend with more passages, but the improvement is not proportional to the retrieval performance. This indicates the limited ability of LLMs in utilizing multiple passages within context.

## 5.4 Other Ablations

We provide additional ablations in §G. In summary, we find that (1) using comprehensive instructions enhances the citation quality of instruction-tuned models (§G.2); (2) including at least one demonstration improves the performance (§G.3); (3) fine-tuned models (FiD; Izacard and Grave, 2021) with POSTCITE lag behind LLMs in both correctness and citation quality and fail to generalize (§G.4).

## 6 Human Evaluation

To verify that our automatic evaluation correlates with human judgement, we conduct human evaluation on selected models and request workers to judge model generations on three dimensions similar to Liu et al. (2023)—(1) utility: a 1-to-5 score indicating whether the generation helps answer the question; (2) citation recall: the annotator is given a sentence and all passages that the sentence cited, and is asked to judge whether the passages fully support the sentence; (3) citation precision: given a sentence and one of its citations, the annotator is asked to judge whether the citation "fully supports", "partially supports", or "does not support" the sentence. Each citation gets a precision score 1 if the output sentence has a citation recall of 1 and this citation at least "partially supports" it. See Appendix F for more details.

**Model outputs score high utility.** The utility scores do not differ significantly between models, ranging 3.7-3.9 for ASQA and 3.5-3.6 for ELI5. Upon inspection, all tested models are mostly able

|  | Human scores | | ALCE scores | |
|---|---|---|---|---|
|  | Rec. | Prec. | Rec. | Prec. |
| ChatGPT Vanilla | 74.7 | 76.6 | 75.3 | 74.4 |
| w/ Rerank | **79.3** | **81.9** | **83.9** | **80.8** |
| Vicuna-13B Vanilla | 51.6 | 51.5 | 50.3 | 50.1 |

Table 8: Human citation quality evaluation vs. ALCE citation quality evaluation on ASQA.

|  | Human scores | | ALCE scores | |
|---|---|---|---|---|
|  | Rec. | Prec. | Rec. | Prec. |
| ChatGPT Vanilla | 50.8 | 52.4 | 52.8 | 50.4 |
| w/ Rerank | **59.7** | **60.6** | **63.0** | **60.6** |
| Vicuna-13B Vanilla | 13.4 | 19.2 | 13.6 | 18.1 |

Table 9: Human citation quality evaluation vs. ALCE citation quality evaluation on ELI5.

to output fluent answers that are related to the question, despite differences in factual correctness.

**Our automatic evaluation of citation quality strongly correlates with human judgements.** As shown in Table 8 (ASQA) and Table 9 (ELI5), the relative rankings induced by human and our automatic metrics are consistent. The absolute citation scores from human and ALCE are very close except for Rerank (which uses the automated citation recall for reranking). This suggests that an improvement on ALCE citation metrics translates to improvement on human preferences. Furthermore, the Cohen's kappa coefficient between human and ALCE suggests substantial agreement for citation recall (0.698) and moderate agreement for citation precision (0.525). We also show in §G.5 that our automatic evaluation achieves high accuracy when treating human annotations as gold labels (85.1% for citation recall and 77.6% for citation precision).

## 7  Related Work

**Evaluating citations.** Generating text with citations is closely related to attribution. Rashkin et al. (2023) define the "attributable to identified sources" (AIS) score to measure how faithful a generated text is to its sources. Bohnet et al. (2022) apply AIS scores on a single-document short-answer QA dataset. Honovich et al. (2022); Yue et al. (2023) study automatic evaluations for the AIS score. A concurrent work (Liu et al., 2023) conduct human evaluation on commercial generative search engines to examine their citation qualities.

Scientific citation text generation (Funkquist et al., 2022) is a related task to ALCE where the

model is provided the papers-to-cite and context and is required to recover the citing text. It is different from ALCE as all citations are provided and the model only needs to perform the summarization.

**Retrieval-augmented LMs.** Many studies have explored augmenting LMs with externally retrieved information. Guu et al. (2020); Borgeaud et al. (2022); Izacard et al. (2022) pre-train language models with retrieved passages, while Khandelwal et al. (2020); Zhong et al. (2022) augment LLMs' output by interpolating it with a $k$NN module; though none of them explicitly provide citations to the retrieved sources. Other works prompt or fine-tune LLMs to "retrieve on-the-fly" (Parisi et al., 2022; Schick et al., 2023; Shuster et al., 2022; Jiang et al., 2023; Yao et al., 2023; Press et al., 2022), which offers flexibility of when and what to search. Gao et al. (2023); He et al. (2022) propose to first generate text without accessing external documents and then retrieve relevant documents and revise the generation to be consistent.

Among previous explorations, Nakano et al. (2021); Menick et al. (2022) are the closest to our setting, where LLMs are trained to answer questions while providing citations. However, they do not explore retrieval strategies and simply use commercial search engines, which are not reproducible, and their models and training data are closed-source. To the best of our knowledge, we are the first to implement end-to-end systems that retrieve, synthesize, and cite documents with LLMs.

## 8  Conclusion

We propose ALCE, the first automatic benchmark for evaluating LLM generations with citations. We deploy automatic metrics to measure fluency, correctness, and citation quality, and verify their efficacy via human evaluation. We explore a variety of strategies for incorporating citations in LLMs and demonstrate that current systems have considerable room for improvement on ALCE.

Our experiments highlight a number of promising research directions, including (1) enhancing retrieval and refining retrieval integrations in LLMs, (2) developing long-context LLMs, and (3) advancing LLMs' ability to synthesize multiple sources. What's even more intriguing is that these research proposals extend beyond the ALCE setup (for example, long-context LLMs have numerous exciting applications), and ALCE can serve as a valuable testbed for their development.

## Limitations

Our evaluation still has room for improvement: (1) MAUVE is found to be sensitive to output length and may provide unstable results; (2) for the ELI5's correctness evaluation, the automatically generated claims may not cover all possible answers due to the open-ended nature of the questions; (3) our citation quality evaluation is limited by the accuracy of the NLI model; for citation precision, the NLI model cannot detect the case of "partially support" and thus leads to a lower citation precision score than the human evaluation.

Although we believe our curated datasets closely resemble the distribution of real-world user questions, we acknowledge that they do not cover more challenging scenarios, such as multi-hop reasoning, math reasoning, and code completion.

In our experiments, we focus on prompting LLMs without updating their model weights. Training a model directly to incorporate citations remains challenging due to the lack of supervised data. However, we observe that certain human-instruction datasets contain examples similar to our task setup. We leave the exploration of training LLMs to generate citations for future work.

## Acknowledgments

We appreciate the helpful feedback from the members of the Princeton NLP group. We thank Alexander Wettig, Nelson Liu, Tianyi Zhang, Yu Meng, Sadhika Malladi, Yangsibo Huang, Zhiyuan Zeng, and Dan Friedman for the valuable discussion. We thank Surge AI (especially Anna Folinsky and Edwin Chen) for their support with the human evaluation. Tianyu Gao is supported by an IBM PhD Fellowship. This research is supported by an NSF CAREER award (IIS-2239290), a Sloan Research Fellowship, and Microsoft Azure credits through the "Accelerate Foundation Models Academic Research" Initiative.

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

## A   Generating Claims for ELI5

We elect not to use ROUGE-L as our main correctness metrics since it does not account for the different ways of expressing the same answer and it can be easily gamed (Krishna et al., 2021). We further illustrate this issue in Table 10. A system can easily achieve high ROUGE-L score by retrieving and returning the top passage from a BM25 index. However, the claims evaluation metric does not reward this approach since the output often lacks different aspects of the answers.

|  | ROUGE-L | Claim recall |
|---|---|---|
| ChatGPT VANILLA | 20.6 | 12.0 |
| ChatGPT ORACLE | 21.2 | 21.3 |
| LLaMa-13B VANILLA | 16.2 | 3.9 |
| Top-1 passage | 19.1 | 3.0 |

Table 10: Comparison between ROUGE-L and claim recall scores on ELI5.

Instead, we leverage the original answers to generate sub-claims and use them to serve as an estimate of the different aspects of the answers that we expect the model to cover. This approach is inspired by works in summarization evaluation and claim verification (Zhang and Bansal, 2021; Kamoi et al., 2023; Wang et al., 2020).

Specifically, we use `text-davinci-003` to generate the sub-claims. We first manually annotate three question and answer pairs from the original ELI5 training set with 3 sub-claims each. Then, we prompt `text-davinci-003` with these pairs as demonstrations. The full prompt with an example is shown in Table 22.

**InstructGPT generates coherent and faithful sub-claims.** To ensure that the generated sub-claims are of good quality, we manually inspect a random sample of 40 answers and their generated sub-claims (totaling to 120 sub-claims). For each sub-claim, we assign a score of 1 if it is relevant to the question and faithful to the facts presented in the ground truth, and 0 otherwise. We found that 112 out of the 120 ($93.33\%$) sub-claims received a score of 1, meaning that our generated sub-claims are of high quality and faithful to the ground truth. Furthermore, the average number of words in the generated sub-claims is 14 words, and they are typically just one sentence long. This is aligned with the intent behind the metric—to capture short factual claims made by the original answer.

**NLI model accurately predicts the entailment of sub-claims.** We further analyze our sub-claim evaluation metrics by checking the error rate of the final prediction of the NLI model. To this end, we first manually annotate the entailment scores between 40 outputs and their sub-claims (in total of 120 pairs; these are the same questions from the previous analysis). We then use the NLI model to obtain the entailment scores for the output and sub-claims. Using the human annotations as the ground truth label, we found that the NLI model achieved an accuracy of $80.0\%$.

## B   Dataset Statistics

For ASQA, human answers have an average length of 65 words. For QAMPARI, each question has on average 13 answers. For ELI5, human answers have an average length of 131 words.

## C   Implementation Details

**NLI model.** We use the version of TRUE model from `https://huggingface.co/google/t5_xxl_true_nli_mixture`, which is trained on SNLI (Bowman et al., 2015), MNLI (Williams et al., 2018), Fever (Thorne et al., 2018), Scitail (Khot et al., 2018), PAWS (Zhang et al., 2019), and VitaminC (Schuster et al., 2021). This model uses the following prompt: "`premise: {PREMISE} hypothesis: {}`" and outputs "1" if the premise entails the hypothesis. We format each passage (when used as premise) by the format of "`Title: {TITLE}\n{TEXT}`" and concatenate all passages with "`\n`" as a separator.

**MAUVE.** When running MAUVE, we concatenate the question and the model output (or human answer) by space. We truncate both the references and the model generations to 100 words, as we found MAUVE results are unstable beyond this length for ELI5 (this is due to that ELI5 has a lot of extremely long human answers).

**Exact match for ASQA and QAMPARI.** Both ASQA and QAMPARI provide aliases for their short answers. We normalize the response and the short answers similarly to Rajpurkar et al. (2016) and report the score with the best-matching aliases. For ASQA, Stelmakh et al. (2022) also propose a QA-based evaluation which we found to be not as stable, and thus we do not report it in our paper.

**Output truncation.** Before evaluation, we trun-

cate model output by new lines, as non-instruction-tuned models may generate more content after new lines that are irrelevant.

**INTERACT.** Empirically, we found that models tend to execute too many consecutive "check" actions, so we force the model to always "output" after each "check". We limit the maximum number of passages to check as 3 to avoid exceeding the length limit. The full passages are removed from the context after one action to save context space. Table 27 provides an example for INTERACT.

**Main experiments.** For all experiments except ChatGPT RERANK, we run each model three times with different seeds and each time we sample two demonstrations from a pool of four. We report the averaged scores for all experiments in the main paper and we report the standard deviations in Appendix G.6.

**Decoding methods.** Based on preliminary experiments we choose the following decoding methods: For ChatGPT and GPT-4, we use sampling with temperature $0.5$; for all open-source models, we use Nucleus sampling (Holtzman et al., 2020) and set top_p $= 0.95$.

## D ALCE Catches Shortcut Cases

| | Fluency | Correct. | Citation | |
| --- | --- | --- | --- | --- |
| | (MAUVE) | (EM Rec.) | Rec. | Prec. |
| ChatGPT | 66.6 | 40.4 | 73.6 | 63.0 |
| Top-1 passage | 20.8 | 35.1 | 99.4 | 99.4 |
| First 2 sents | 67.2 | 18.9 | 98.7 | 98.7 |

Table 11: ASQA cheating cases. "ChatGPT": the ChatGPT VANILLA model with GTR-retrieved top-5 passages. "Top-1 passage": use the top-1 retrieved passage as the response. "First 2 sents": use the first 2 sentences of the top-1 retrieved passage.

Table 11 demonstrates the experiments to show that ALCE is robust to shortcut cases. Using the top-1 passages or first two sentences of the top-1 passages induces almost perfect citation quality, but fluency and correctness are dramatically lower.

## E Citation Recall Discussion

Our citation precision evaluation cannot detect a citation that partially supports the statement and hence will falsely penalize it. Consider a statement $s_3$ and its citations [2][4][5]: if [2] entails partial information of $s_3$ that [4][5] also entails,

[2] will be counted as "irrelevant" while it should not be penalized. Liu et al. (2023) conduct human evaluation on citation precision in a different way: For each citation, they ask annotators to judge whether the citation (1) fully support, (2) partially support, or (3) does not support $s_i$. One citation $c_{i,j}$ is precise if (a) $c_{i,j}$ fully supports $s_i$ or (b) $\mathcal{C}_i$ fully supports $s_i$, $c_{i,j}$ partially supports $s_i$, and no $c \in \mathcal{C}_i$ alone fully supports $s_i$. This evaluation solved the corner case we mentioned in the main paper (one citation partially supports the claim but is identified as "irrelevant"). However, it is challenging to conduct such evaluation automatically, as there is no existing model that can judge whether a citation "partially" supports a claim. We also explore prompting ChatGPT to conduct such a task, which yields poor results. We defer it to future work to collect supervised data to train a better $\phi$ that can detect "partial support".

## F Human Evaluation

We employ Surge AI (https://www.surgehq.ai/) for our human evaluation. The average pay to workers is 20 USD per hour. We randomly sample 100 examples from ASQA and ELI5 and annotate outputs of selected models: ChatGPT VANILLA, ChatGPT RERANK, and Vicuna-13B VANILLA.

### F.1 Utility

To check if the model output is useful to downstream users, we measure the utility of the response $\mathcal{S}$. We first show the query $q$ and model response $\mathcal{S}$ to the worker and ask them to rate their agreement with the statement "The response is a helpful and informative answer to the query" on a Likert scale of 1-5, corresponding to *Strongly Disagree*, *Disagree*, *Neutral*, *Agree*, and *Strongly Agree*.

### F.2 Citation Recall

The annotators are shown the question $q$, the statement $s_i$, and all of its citations $\mathcal{C}_i$, and they rate if the joint set of citations fully support the statement (recall=1) or if they do not support all the claims (recall=0). We calculate the overall recall score for the generation by taking an average of all the statements' recall scores.

### F.3 Citation Precision

We show the question $q$ and a pair of a statement $s_i$ and one of its citation $c_{i,j} \in \mathcal{C}_i$ to the annotator. We ask the annotator if the citation *fully supports*,

|        | R@1  | R@3  | R@5  | R@20 | R@100 |
|--------|------|------|------|------|-------|
| DPR    | 29.6 | 44.5 | 51.5 | 64.6 | 74.1  |
| GTR    | 35.1 | 50.7 | 56.8 | 70.3 | 78.4  |
| Oracle | 63.8 | 72.8 | 78.4 | -    | -     |

Table 12: Retrieval results for ASQA (EM recall).

|        | R@1  | R@3  | R@5  | R@20 | R@100 |
|--------|------|------|------|------|-------|
| DPR    | 6.7  | 13.5 | 17.6 | 30.4 | 47.6  |
| GTR    | 14.6 | 24.7 | 31.6 | 49.7 | 65.6  |
| Oracle | 44.3 | 58.7 | 65.6 | -    | -     |

Table 13: Retrieval results for QAMPARI (recall-5).

|        | R@1  | R@3  | R@5  | R@20 | R@100 |
|--------|------|------|------|------|-------|
| BM25   | 3.0  | 6.6  | 9.6  | 19.3 | 31.8  |
| Oracle | 25.3 | 29.7 | 31.8 | -    | -     |

Table 14: Retrieval results for ELI5 (claim recall).

|                   | Fluency  | Correct.  | Citation |       |
|-------------------|----------|-----------|----------|-------|
|                   | (MAUVE)  | (EM Rec.) | Rec.     | Prec. |
| **ChatGPT** (VANILLA, 5-doc) | | | | |
| Short instruction | 64.1     | 39.5      | 69.6     | 73.2  |
| Full instruction  | 66.6     | 40.4      | 73.6     | 72.5  |

Table 15: Effect of different instructions on ASQA.

*partially supports*, or *does not support* the factual claims in $s_i$. Citation $c_{i,j}$ has a citation precision of 1 if $s_i$ has a recall of 1, and $c_{i,j}$ fully or partially supports $s_i$. Finally, we take an average of precision scores of all citations in the statement $\mathcal{S}$ to obtain the citation precision score.

# G   More Experiments

## G.1   Retrieval Analysis

**Oracle.** Since the original datasets do not contain gold passages at the same granularity level as our setting (100-word passages), we approximate gold passages by running the following algorithm on the top-100 retrieved passages. We first calculate the recall score for each passage. Then, we sort the passages using their recall score and take the top 5 passages as our initial oracle set. Finally, we iterate through all passages that were not initially in the oracle set and try to replace the passages in the oracle set in a greedy fashion: we calculate the change in the recall score of the oracle set for every possible replacement and proceed with the replacement that results in the largest recall improvement. The set of 5 oracle passages were able to match the recall scores of the top-100 retrieved passages.

**Detailed retrieval results.** We show detailed retrieval results in Tables 12, 13, and 14.

## G.2   Effect of Instructions

Table 15 shows results of using a full instruction (Table 23) and a short version of the instruction (Table 24). We see that the full version induces stronger correctness and citation recall, while the two instructions lead to similar citation precision.

## G.3   Effect of Demonstrations

Table 16 shows results on effect of different numbers of demonstrations. We see that numbers of demonstrations do not affect ChatGPT's correctness but using at least one demonstration ensures high citation recall. For the original LLaMA model, Table 16 shows the trend that more demonstrations lead to better performance.

## G.4   Fine-tuned Models

To better understand the differences between fine-tuned models and prompted large language models, we train state-of-the-art question answering model, Fusion-in-Decoder (FiD; Izacard and Grave (2021)), and evaluate it in conjunction with POSTCITE. Due to the lack of training data with citation annotation, we first train a T5-base FiD model for 5 epochs on the ASQA training set with a batch size of 64 and a learning rate of 1e-4. During evaluation, we use POSTCITE to add citations to the output. We also use $k = 5$ passages during both training and evaluation of the FiD model.

Then, we evaluate this model on both ASQA (in-domain) and ELI5 (out-of-domain), and the results can be found in Tables 17 and 18. Note that this is not a direct comparison, as ALCE assumes only evaluation data available and uses only few-shot data for prompting. As the results show, the FiD baseline still significantly lags behind prompting ChatGPT in both correctness and citation quality (even though it is trained on 4000+ examples). When tested on another dataset (ELI5), FiD performs even worse, showing that it is challenging to solve the problem by fine-tuning a small pre-trained model.

| | Fluency | Correct. | Citation | |
|---|---|---|---|---|
| | (MAUVE) | (EM) | Rec. | Prec. |
| **ChatGPT** (Vanilla) | | | | |
| #demo = 0 | 74.5 | 41.9 | 69.3 | 73.4 |
| #demo = 1 | 68.9 | 39.8 | 74.6 | 73.2 |
| #demo = 2 | 66.6 | 40.4 | 73.6 | 72.5 |

Table 16: Different demonstrations on ASQA.

| | Fluency | Correct. | Citation | |
|---|---|---|---|---|
| | (MAUVE) | (EM Rec.) | Rec. | Prec. |
| ChatGPT | 66.6 | 40.4 | 73.6 | 72.5 |
| FiD + PostCite | 75.8 | 28.4 | 58.1 | 58.0 |

Table 17: Comparison of Fusion-in-Decoder with Chat-GPT on ASQA. Both models use top-5 GTR passages.

## G.5 More Human Evaluation

We evaluate the accuracy of our automatic metrics by treating the human annotations as gold labels. For citation recall, ALCE achieves an accuracy of $85.1\%$; for citation precision, ALCE has an accuracy of $77.6\%$. Regarding detecting insufficient citations, ALCE has a recall of $82.3\%$ and a precision of $84.2\%$; regarding detecting "irrelevant" citations, ALCE has a recall of $75.6\%$ and a precision of $66.1\%$—ALCE is effective in detecting "irrelevant" citations, but due to the limitation of the NLI model (cannot detect "partial support"), it has a relatively high false positive rate.

## G.6 Main Results

We show full results of our experiments along with the standard deviation in Tables 19, 20, and 21. We repeat all experiments with three different random seeds. However, for ChatGPT Rerank, we use only one seeded run since each run repeats the generation step four times, and more experiments would incur significant costs.

## H Prompts

We show detailed prompts used in our paper in Tables 23, 24, 25, 26, 27, 28, and 29.

## I Examples

In Tables 30 and 31 we show some examples of questions and model generated outputs.

| | Fluency | Correct. | Citation | |
|---|---|---|---|---|
| | (MAUVE) | (Claim) | Rec. | Prec. |
| ChatGPT | 57.2 | 12.0 | 51.1 | 50.0 |
| FiD + PostCite | 25.2 | 4.4 | 39.3 | 39.3 |

Table 18: Comparison of Fusion-in-Decoder with Chat-GPT on ELI5. Both models use top-5 GTR passages.

| | Fluency | Correct. | Citation | | ROUGE-L | Length |
|---|---|---|---|---|---|---|
| | (MAUVE) | (EM Rec.) | Rec. | Prec. | | |
| **ChatGPT** | | | | | | |
| VANILLA (5-psg) | 66.8 (2.0) | 40.4 (0.6) | 73.6 (1.1) | 72.5 (1.8) | 37.0 (0.4) | 40.0 (3.1) |
| w/ RERANK | 77.0 (0.0) | 40.2 (0.0) | 84.8 (0.0) | 81.6 (0.0) | 36.9 (0.0) | 40.8 (0.0) |
| SUMM (10-psg) | 70.0 (1.2) | 43.3 (0.8) | 68.8 (0.6) | 61.8 (1.1) | 36.9 (0.2) | 49.8 (4.3) |
| w/ INTERACT | 69.0 (2.7) | 39.1 (0.5) | 73.4 (0.2) | 66.5 (4.9) | 35.7 (0.2) | 34.0 (0.9) |
| SNIPPET (10-psg) | 69.8 (2.5) | 41.4 (0.6) | 65.3 (0.6) | 57.4 (0.9) | 36.4 (0.4) | 43.0 (3.5) |
| INLINESEARCH | 58.7 (1.3) | 32.4 (0.6) | 58.3 (1.3) | 58.3 (1.3) | 58.2 (1.1) | 23.7 (1.1) |
| CLOSEDBOOK | 52.7 (4.9) | 38.2 (0.1) | 26.7 (1.1) | 26.7 (1.1) | 37.1 (0.3) | 61.1 (4.5) |
| ORACLE(5-psg) | 64.4 (0.6) | 48.9 (1.2) | 74.5 (0.6) | 72.7 (1.0) | 38.2 (1.0) | 37.4 (3.0) |
| **ChatGPT-16K** | | | | | | |
| VANILLA (5-psg) | 60.3 (−) | 36.1 (−) | 76.2 (−) | 76.5 (−) | 36.2 (−) | 24.7 (−) |
| VANILLA (10-psg) | 56.3 (−) | 36.7 (−) | 75.3 (−) | 75.0 (−) | 35.6 (−) | 23.5 (−) |
| VANILLA (20-psg) | 56.7 (−) | 36.1 (−) | 73.7 (−) | 73.5 (−) | 35.5 (−) | 23.1 (−) |
| **GPT-4** | | | | | | |
| VANILLA (5-psg) | 67.1 (−) | 41.3 (−) | 68.5 (−) | 75.6 (−) | 39.2 (−) | 31.8 (−) |
| VANILLA (10-psg) | 71.5 (−) | 43.1 (−) | 72.0 (−) | 75.5 (−) | 39.7 (−) | 33.8 (−) |
| VANILLA (20-psg) | 64.9 (−) | 44.4 (−) | 73.0 (−) | 76.5 (−) | 40.1 (−) | 34.3 (−) |
| **Open-source** | | | | | | |
| LLaMA-7B VANILLA (3-psg) | 69.8 (2.0) | 22.6 (0.9) | 6.2 (2.7) | 9.2 (2.9) | 29.1 (0.2) | 61.3 (14.3) |
| Alpaca-7B VANILLA (3-psg) | 84.2 (2.7) | 32.1 (1.7) | 12.3 (7.2) | 14.1 (7.0) | 33.1 (0.8) | 51.7 (12.8) |
| Vicuna-7B VANILLA (3-psg) | 82.9 (5.0) | 34.6 (0.7) | 40.3 (0.5) | 42.6 (1.0) | 35.9 (0.7) | 48.9 (6.6) |
| LLaMA-13B VANILLA (3-psg) | 68.4 (6.4) | 26.9 (0.4) | 10.6 (4.7) | 15.4 (5.2) | 29.8 (0.5) | 67.1 (19.1) |
| w/ RERANK | 60.9 (14.5) | 25.2 (2.5) | 28.1 (9.3) | 37.0 (7.2) | 27.9 (2.4) | 50.5 (14.3) |
| LLaMA-13B SUMM (10-psg) | 76.8 (4.7) | 33.3 (0.7) | 19.6 (3.9) | 23.7 (4.7) | 32.1 (0.3) | 54.4 (1.5) |
| LLaMA-13B SNIPPET (10-psg) | 72.0 (0.8) | 31.3 (1.1) | 18.2 (3.1) | 21.1 (3.6) | 30.8 (0.4) | 50.5 (4.5) |
| LLaMA-13B ORACLE (3-psg) | 69.5 (11.4) | 34.3 (0.9) | 10.8 (4.9) | 15.8 (5.9) | 30.6 (0.1) | 67.3 (17.9) |
| Vicuna-13B VANILLA (3-psg) | 82.6 (9.4) | 31.9 (3.9) | 51.1 (1.4) | 50.1 (2.5) | 34.9 (1.3) | 39.1 (6.6) |
| w/ RERANK | 73.5 (2.1) | 32.9 (1.3) | 71.9 (1.9) | 65.4 (1.5) | 34.6 (0.3) | 35.7 (4.2) |
| Vicuna-13B SUMM (10-psg) | 67.7 (0.3) | 43.2 (0.1) | 52.7 (2.6) | 50.0 (2.1) | 36.7 (0.2) | 66.0 (1.2) |
| Vicuna-13B SNIPPET (10-psg) | 81.4 (3.0) | 42.1 (1.2) | 53.4 (1.9) | 48.7 (1.6) | 36.9 (0.4) | 61.2 (7.4) |
| Vicuna-13B ORACLE (3-psg) | 72.9 (3.5) | 42.5 (1.6) | 52.2 (0.8) | 50.7 (1.6) | 36.5 (0.9) | 38.7 (3.5) |
| LLaMA-33B VANILLA (3-psg) | 83.7 (5.4) | 31.0 (0.8) | 19.5 (5.3) | 23.0 (5.3) | 32.3 (0.6) | 44.1 (9.3) |
| w/ RERANK | 82.1 (3.0) | 31.3 (1.1) | 41.3 (6.4) | 44.7 (5.5) | 32.5 (0.9) | 39.4 (8.0) |
| LLaMA-33B SUMM (10-psg) | 72.0 (3.0) | 33.1 (1.9) | 34.7 (5.8) | 35.2 (6.0) | 31.1 (0.8) | 43.7 (5.0) |
| LLaMA-33B SNIPPET (10-psg) | 70.8 (3.1) | 30.9 (1.4) | 31.4 (4.2) | 31.5 (5.3) | 30.1 (0.7) | 42.8 (3.6) |
| LLaMA-33B ORACLE (3-psg) | 82.6 (7.1) | 39.3 (2.9) | 20.2 (6.2) | 23.9 (6.3) | 33.1 (0.9) | 42.0 (9.3) |
| Oasst-33B VANILLA (3-psg) | 82.9 (2.7) | 34.8 (1.5) | 36.2 (1.7) | 38.3 (2.7) | 35.5 (0.7) | 45.2 (6.3) |
| w/ RERANK | 83.2 (2.4) | 35.1 (1.4) | 66.7 (0.2) | 64.3 (1.0) | 35.0 (0.6) | 41.8 (6.0) |
| Oasst-33B SUMM (10-psg) | 74.3 (4.6) | 40.9 (1.1) | 45.5 (1.9) | 44.0 (2.9) | 35.8 (0.6) | 54.3 (4.8) |
| Oasst-33B SNIPPET (10-psg) | 79.3 (1.0) | 40.1 (0.9) | 45.0 (1.3) | 43.3 (2.2) | 35.8 (0.2) | 50.9 (4.1) |
| Oasst-33B ORACLE (3-psg) | 85.1 (2.8) | 44.3 (2.4) | 37.0 (1.0) | 39.6 (1.5) | 36.5 (1.1) | 44.2 (5.8) |
| LLaMA-2-7B-Chat VANILLA (5-psg) | 80.1 (6.5) | 33.9 (2.1) | 50.9 (4.5) | 47.5 (3.7) | 35.1 (0.9) | 42.3 (10.1) |
| LLaMA-2-13B-Chat VANILLA (5-psg) | 72.4 (6.3) | 35.2 (1.2) | 38.4 (5.9) | 39.4 (4.8) | 35.8 (0.9) | 38.0 (6.4) |
| LLaMA-2-70B-Chat VANILLA (5-psg) | 88.3 (4.1) | 41.5 (0.8) | 62.9 (1.4) | 61.3 (2.1) | 37.1 (0.4) | 52.9 (9.5) |

Table 19: ASQA full results.

|  | Correctness | | Citation | | |
| --- | --- | --- | --- | --- | --- |
|  | Rec.-5 | Prec. | Rec. | Prec. | Num Pred. |
| **ChatGPT** | | | | | |
| VANILLA (5-psg) | 20.8 $_{(2.2)}$ | 20.8 $_{(0.2)}$ | 20.5 $_{(0.7)}$ | 20.9 $_{(0.7)}$ | 5.0 $_{(0.5)}$ |
| w/ RERANK | 22.8 $_{(0.0)}$ | 21.4 $_{(0.0)}$ | 21.2 $_{(0.0)}$ | 21.4 $_{(0.0)}$ | 5.4 $_{(0.0)}$ |
| SUMM (10-psg) | 23.6 $_{(0.9)}$ | 21.2 $_{(0.5)}$ | 23.6 $_{(0.7)}$ | 25.7 $_{(0.8)}$ | 6.7 $_{(0.4)}$ |
| SNIPPET (10-psg) | 24.5 $_{(1.4)}$ | 21.5 $_{(1.8)}$ | 22.9 $_{(1.6)}$ | 24.9 $_{(0.4)}$ | 7.2 $_{(0.9)}$ |
| w/ INTERACT | 21.9 $_{(0.9)}$ | 23.0 $_{(0.4)}$ | 21.9 $_{(1.2)}$ | 23.4 $_{(0.9)}$ | 6.7 $_{(0.4)}$ |
| INLINESEARCH | 17.2 $_{(1.1)}$ | 20.4 $_{(0.8)}$ | 14.9 $_{(0.8)}$ | 14.9 $_{(0.8)}$ | 6.7 $_{(0.2)}$ |
| CLOSEDBOOK | 32.9 $_{(1.1)}$ | 19.8 $_{(1.6)}$ | 10.0 $_{(0.4)}$ | 10.0 $_{(0.4)}$ | 17.0 $_{(2.9)}$ |
| ORACLE | 37.0 $_{(3.1)}$ | 36.9 $_{(0.6)}$ | 24.1 $_{(1.2)}$ | 24.6 $_{(1.3)}$ | 5.3 $_{(0.6)}$ |
| **ChatGPT-16K** | | | | | |
| VANILLA (5-psg) | 21.1 $_{(-)}$ | 22.0 $_{(-)}$ | 20.7 $_{(-)}$ | 21.2 $_{(-)}$ | 4.9 $_{(-)}$ |
| VANILLA (10-psg) | 23.4 $_{(-)}$ | 21.9 $_{(-)}$ | 21.6 $_{(-)}$ | 22.0 $_{(-)}$ | 5.7 $_{(-)}$ |
| VANILLA (20-psg) | 26.4 $_{(-)}$ | 21.1 $_{(-)}$ | 19.4 $_{(-)}$ | 19.7 $_{(-)}$ | 7.6 $_{(-)}$ |
| **GPT-4** | | | | | |
| VANILLA (5-psg) | 22.2 $_{(-)}$ | 25.0 $_{(-)}$ | 25.9 $_{(-)}$ | 27.0 $_{(-)}$ | 4.4 $_{(-)}$ |
| VANILLA (10-psg) | 26.8 $_{(-)}$ | 25.1 $_{(-)}$ | 26.2 $_{(-)}$ | 27.2 $_{(-)}$ | 5.7 $_{(-)}$ |
| VANILLA (20-psg) | 29.6 $_{(-)}$ | 26.2 $_{(-)}$ | 27.4 $_{(-)}$ | 28.5 $_{(-)}$ | 6.8 $_{(-)}$ |
| **Open-source** | | | | | |
| LLaMA-7B VANILLA (3-psg) | 7.8 $_{(3.4)}$ | 7.4 $_{(2.7)}$ | 5.1 $_{(0.5)}$ | 5.7 $_{(0.8)}$ | 5.7 $_{(0.6)}$ |
| Alpaca-7B VANILLA (3-psg) | 9.4 $_{(3.7)}$ | 9.5 $_{(3.6)}$ | 6.4 $_{(0.5)}$ | 6.8 $_{(0.5)}$ | 5.1 $_{(0.1)}$ |
| Vicuna-7B VANILLA (3-psg) | 11.3 $_{(1.4)}$ | 13.3 $_{(2.3)}$ | 10.1 $_{(0.6)}$ | 10.9 $_{(0.5)}$ | 3.9 $_{(0.3)}$ |
| LLaMA-13B VANILLA (3-psg) | 9.7 $_{(3.6)}$ | 9.1 $_{(3.1)}$ | 6.7 $_{(0.9)}$ | 7.1 $_{(0.9)}$ | 5.9 $_{(0.6)}$ |
| w/ RERANK | 10.0 $_{(3.3)}$ | 10.7 $_{(3.3)}$ | 9.9 $_{(1.2)}$ | 10.2 $_{(1.1)}$ | 5.4 $_{(0.5)}$ |
| LLaMA-13B SUMM (10-psg) | 14.8 $_{(2.5)}$ | 12.6 $_{(1.5)}$ | 7.4 $_{(0.5)}$ | 8.0 $_{(0.6)}$ | 8.1 $_{(0.9)}$ |
| LLaMA-13B SNIPPET (10-psg) | 17.7 $_{(1.4)}$ | 15.7 $_{(0.9)}$ | 8.8 $_{(0.7)}$ | 9.9 $_{(0.6)}$ | 8.2 $_{(0.4)}$ |
| LLaMA-13B ORACLE (3-psg) | 16.8 $_{(6.6)}$ | 15.4 $_{(5.6)}$ | 7.7 $_{(1.0)}$ | 8.3 $_{(1.1)}$ | 5.7 $_{(0.7)}$ |
| Vicuna-13B VANILLA (3-psg) | 14.0 $_{(0.6)}$ | 15.9 $_{(1.7)}$ | 12.5 $_{(0.8)}$ | 13.4 $_{(0.7)}$ | 4.7 $_{(0.3)}$ |
| w/ RERANK | 13.0 $_{(0.7)}$ | 17.2 $_{(2.2)}$ | 17.3 $_{(0.8)}$ | 17.7 $_{(0.6)}$ | 4.4 $_{(0.3)}$ |
| Vicuna-13B SUMM (10-psg) | 21.1 $_{(1.4)}$ | 17.1 $_{(0.3)}$ | 15.7 $_{(0.2)}$ | 17.8 $_{(0.1)}$ | 6.9 $_{(0.7)}$ |
| Vicuna-13B SNIPPET (10-psg) | 21.9 $_{(0.8)}$ | 18.2 $_{(0.3)}$ | 16.8 $_{(0.3)}$ | 19.7 $_{(0.6)}$ | 7.5 $_{(0.4)}$ |
| Vicuna-13B ORACLE (3-psg) | 25.9 $_{(1.6)}$ | 28.4 $_{(2.6)}$ | 15.8 $_{(1.4)}$ | 16.8 $_{(1.4)}$ | 4.9 $_{(0.5)}$ |
| LLaMA-33B VANILLA (3-psg) | 14.7 $_{(3.3)}$ | 12.0 $_{(2.2)}$ | 7.9 $_{(0.7)}$ | 8.3 $_{(0.6)}$ | 7.2 $_{(0.7)}$ |
| w/ RERANK | 14.0 $_{(3.4)}$ | 13.9 $_{(2.6)}$ | 10.7 $_{(0.6)}$ | 11.1 $_{(0.5)}$ | 6.4 $_{(0.7)}$ |
| LLaMA-33B SUMM (10-psg) | 19.0 $_{(1.9)}$ | 14.8 $_{(0.8)}$ | 12.5 $_{(0.2)}$ | 15.0 $_{(0.3)}$ | 7.6 $_{(0.6)}$ |
| LLaMA-33B SNIPPET (10-psg) | 19.6 $_{(1.1)}$ | 15.7 $_{(0.1)}$ | 12.8 $_{(1.1)}$ | 15.2 $_{(1.2)}$ | 7.8 $_{(0.5)}$ |
| LLaMA-33B ORACLE (3-psg) | 23.9 $_{(6.9)}$ | 20.3 $_{(5.2)}$ | 9.8 $_{(1.2)}$ | 10.4 $_{(1.2)}$ | 6.8 $_{(0.9)}$ |
| Oasst-33B VANILLA (3-psg) | 15.5 $_{(1.5)}$ | 14.9 $_{(1.4)}$ | 9.0 $_{(1.6)}$ | 10.1 $_{(1.8)}$ | 5.6 $_{(0.3)}$ |
| w/ RERANK | 14.1 $_{(1.1)}$ | 15.8 $_{(1.0)}$ | 15.0 $_{(1.6)}$ | 15.9 $_{(1.6)}$ | 4.7 $_{(0.3)}$ |
| Oasst-33B SUMM (10-psg) | 21.0 $_{(0.6)}$ | 17.5 $_{(1.0)}$ | 12.9 $_{(1.2)}$ | 16.6 $_{(1.2)}$ | 7.1 $_{(0.4)}$ |
| Oasst-33B SNIPPET (10-psg) | 22.0 $_{(0.4)}$ | 17.4 $_{(0.3)}$ | 13.6 $_{(1.7)}$ | 17.7 $_{(1.6)}$ | 7.5 $_{(0.1)}$ |
| Oasst-33B ORACLE (3-psg) | 26.9 $_{(3.7)}$ | 26.0 $_{(3.3)}$ | 11.7 $_{(1.0)}$ | 12.9 $_{(1.2)}$ | 5.6 $_{(0.4)}$ |
| LLaMA-2-7B-Chat VANILLA (5-psg) | 16.2 $_{(1.3)}$ | 15.3 $_{(1.6)}$ | 10.6 $_{(0.9)}$ | 10.9 $_{(1.0)}$ | 5.5 $_{(0.0)}$ |
| LLaMA-2-13B-Chat VANILLA (5-psg) | 21.1 $_{(0.9)}$ | 18.2 $_{(0.5)}$ | 9.6 $_{(1.5)}$ | 9.7 $_{(1.5)}$ | 6.5 $_{(0.3)}$ |
| LLaMA-2-70B-Chat VANILLA (5-psg) | 21.8 $_{(0.7)}$ | 18.4 $_{(0.1)}$ | 15.1 $_{(1.2)}$ | 15.6 $_{(1.3)}$ | 7.1 $_{(0.2)}$ |

Table 20: QAMPARI full results.

| | Fluency | Correct. | Citation | | ROUGE-L | Length |
|---|---|---|---|---|---|---|
| | (MAUVE) | (Claim) | Rec. | Prec. | | |
| **ChatGPT** | | | | | | |
| VANILLA (5-psg) | 57.2 (1.6) | 12.0 (0.6) | 51.1 (4.2) | 50.0 (4.8) | 20.6 (0.2) | 91.5 (6.5) |
| w/ RERANK | 56.1 (0.0) | 11.4 (0.0) | 69.3 (0.0) | 67.8 (0.0) | 20.3 (0.0) | 103.4 (0.0) |
| SUMM (10-psg) | 40.2 (1.2) | 12.5 (0.2) | 51.5 (1.1) | 48.2 (2.0) | 20.3 (0.1) | 90.0 (6.6) |
| SNIPPET (10-psg) | 62.9 (2.2) | 14.3 (0.1) | 50.4 (1.1) | 45.0 (2.w) | 21.0 (0.1) | 100.0 (6.8) |
| w/ INTERACT | 68.0 (5.8) | 13.3 (0.2) | 47.8 (3.3) | 45.0 (3.1) | 20.1 (0.2) | 99.8 (6.1) |
| INLINESEARCH | 49.7 (4.6) | 13.4 (1.1) | 45.6 (2.5) | 43.7 (3.9) | 20.4 (0.3) | 103.0 (18.1) |
| CLOSEDBOOK | 32.6 (1.1) | 18.6 (0.5) | 15.4 (0.3) | 15.4 (0.3) | 22.8 (0.1) | 108.3 (8.9) |
| ORACLE (5-psg) | 59.4 (4.1) | 21.3 (0.2) | 57.8 (3.7) | 56.0 (3.8) | 21.2 (0.3) | 93.0 (7.8) |
| **ChatGPT-16K** | | | | | | |
| VANILLA (5-psg) | 31.6 (−) | 14.4 (−) | 44.6 (−) | 44.1 (−) | 21.4 (−) | 87.6 (−) |
| VANILLA (10-psg) | 26.6 (−) | 14.4 (−) | 45.5 (−) | 43.3 (−) | 21.5 (−) | 87.5 (−) |
| VANILLA (20-psg) | 31.6 (−) | 15.9 (−) | 43.4 (−) | 40.9 (−) | 21.7 (−) | 92.6 (−) |
| **GPT-4** | | | | | | |
| VANILLA (5-psg) | 38.4 (−) | 14.2 (−) | 44.0 (−) | 50.1 (−) | 20.6 (−) | 79.6 (−) |
| VANILLA (10-psg) | 39.9 (−) | 15.7 (−) | 49.5 (−) | 54.2 (−) | 21.2 (−) | 88.2 (−) |
| VANILLA (20-psg) | 41.5 (−) | 18.3 (−) | 48.5 (−) | 53.4 (−) | 22.2 (−) | 97.0 (−) |
| **Open-source** | | | | | | |
| LLaMA-7B VANILLA (3-psg) | 28.6 (17.9) | 1.6 (0.9) | 1.2 (0.0) | 2.7 (0.1) | 12.2 (1.3) | 46.9 (1.2) |
| Alpaca-7B VANILLA (3-psg) | 45.9 (5.3) | 9.2 (0.1) | 4.5 (1.6) | 5.2 (1.9) | 18.8 (0.3) | 67.1 (1.2) |
| Vicuna-7B VANILLA (3-psg) | 43.2 (3.9) | 10.0 (0.5) | 12.6 (2.3) | 16.3 (2.6) | 19.1 (0.4) | 68.7 (2.0) |
| LLaMA-13B VANILLA (3-psg) | 50.0 (2.0) | 3.9 (0.4) | 3.1 (0.9) | 5.3 (1.3) | 16.1 (0.5) | 63.3 (2.0) |
| w/ RERANK | 46.7 (2.9) | 4.3 (0.4) | 9.7 (2.1) | 15.0 (2.2) | 16.1 (0.7) | 63.0 (2.3) |
| LLaMA-13B SUMM (10-psg) | 28.6 (1.8) | 2.9 (0.1) | 2.5 (0.8) | 3.8 (0.8) | 8.5 (0.3) | 33.1 (0.6) |
| LLaMA-13B SNIPPET (10-psg) | 48.4 (3.1) | 5.7 (0.9) | 5.8 (0.6) | 7.6 (0.9) | 15.1 (1.1) | 60.2 (3.2) |
| LLaMA-13B ORACLE (3-psg) | 49.5 (2.4) | 6.4 (0.6) | 3.7 (0.7) | 6.5 (1.0) | 16.8 (0.5) | 64.5 (1.7) |
| Vicuna-13B VANILLA (3-psg) | 58.2 (25.1) | 10.0 (0.3) | 15.6 (2.2) | 19.6 (2.0) | 19.1 (0.3) | 69.6 (0.6) |
| w/ RERANK | 45.9 (4.3) | 9.2 (0.0) | 31.7 (2.9) | 38.2 (1.6) | 18.6 (0.5) | 69.7 (1.0) |
| Vicuna-13B SUMM (10-psg) | 22.4 (3.0) | 4.9 (0.1) | 9.7 (1.3) | 12.2 (1.2) | 9.3 (0.4) | 33.0 (3.7) |
| Vicuna-13B SNIPPET (10-psg) | 48.1 (5.3) | 11.2 (1.4) | 27.2 (3.6) | 27.9 (1.9) | 18.4 (1.9) | 76.8 (8.7) |
| Vicuna-13B ORACLE (3-psg) | 41.6 (3.1) | 17.1 (0.4) | 20.2 (3.0) | 26.5 (3.0) | 20.0 (0.3) | 72.0 (0.3) |
| LLaMA-33B VANILLA (3-psg) | 58.8 (4.3) | 6.2 (0.0) | 9.3 (3.0) | 12.1 (4.2) | 16.9 (0.2) | 60.0 (1.3) |
| w/ RERANK | 65.9 (2.5) | 6.0 (0.7) | 22.5 (5.2) | 26.1 (6.9) | 17.5 (0.4) | 61.0 (1.2) |
| LLaMA-33B SUMM (10-psg) | 23.3 (2.0) | 3.0 (0.2) | 6.2 (0.5) | 8.2 (0.7) | 7.5 (0.4) | 26.2 (2.3) |
| LLaMA-33B SNIPPET (10-psg) | 53.2 (4.0) | 7.4 (1.3) | 13.7 (0.5) | 15.1 (0.4) | 14.4 (1.7) | 53.3 (8.5) |
| LLaMA-33B ORACLE (3-psg) | 63.7 (2.8) | 11.4 (0.5) | 11.9 (2.6) | 15.4 (3.6) | 17.9 (0.2) | 61.7 (2.6) |
| Oasst-33B VANILLA (3-psg) | 46.8 (7.6) | 9.5 (0.2) | 16.0 (2.5) | 21.6 (3.5) | 18.6 (0.3) | 67.8 (1.5) |
| w/ RERANK | 52.1 (6.1) | 8.5 (0.5) | 34.4 (2.9) | 41.5 (2.5) | 18.2 (0.3) | 67.0 (1.5) |
| Oasst-33B SUMM (10-psg) | 24.8 (2.8) | 3.9 (0.3) | 12.3 (0.2) | 16.3 (0.3) | 9.1 (0.3) | 31.6 (1.5) |
| Oasst-33B SNIPPET (10-psg) | 50.7 (4.6) | 10.7 (1.2) | 25.8 (3.3) | 26.7 (2.3) | 17.8 (1.8) | 69.6 (8.6) |
| Oasst-33B ORACLE (3-psg) | 50.7 (12.1) | 15.8 (0.1) | 20.8 (2.8) | 28.0 (3.2) | 19.4 (0.1) | 70.3 (1.1) |
| LLaMA-2-7B-Chat VANILLA (5-psg) | 27.8 (3.0) | 10.9 (0.2) | 19.8 (1.2) | 15.0 (1.4) | 20.5 (0.2) | 87.8 (8.1) |
| LLaMA-2-13B-Chat VANILLA (5-psg) | 34.7 (1.5) | 13.4 (0.4) | 17.3 (1.3) | 15.8 (1.4) | 20.9 (0.2) | 88.3 (6.3) |
| LLaMA-2-70B-Chat VANILLA (5-psg) | 38.6 (4.8) | 12.8 (1.0) | 38.3 (2.4) | 37.9 (1.9) | 21.3 (0.1) | 110.8 (5.6) |

Table 21: ELI5 full results.

Read the original question and passage, and generate 3 additional claims that are supported by the
passage and answer the question.

Original question: What's the difference between Shia vs. Sunni Islam?
Passage: The main difference between Shia and Sunni Muslim is related to ideological heritage and
issues of leadership. This difference is first formed after the death of the Prophet Muhammad in 632
A.D. The ideological practice of the Sunni branch strictly follows Prophet Muhammad and his
teachings, while the Shia branch follows Prophet Muhammad's son-in-law Ali. Nowadays, Sunni and
Shia are the major branches of Islam.
Claim 1: The major branches of Islam are Sunni and Shia.
Claim 2: Prophet Muhammad died in 632 A.D.
Claim 3: The ideological practice of the Sunni branch strictly follows Prophet Muhammad and his
teachings.

Original question: What causes Bi-polar disorder?
Passage: Bipolar disorder is an emotional disorder that causes extreme mood swings between
excitement and depression. The spectrum of mood swing may span from days to months. We are still not
certain of the exact factors that cause such disorder, but genetics is considered a major factor.
Claim 1: One symptom of Bi-polar disorder is extreme mood swings between excitement and depression.
Claim 2: Genetics could be one of the major factors that causes Bi-polar disorder.
Claim 3: The mood swing from Bi-polar disorder can last days to months.

Original question: How do we hear differences in sound besides volume and pitch?
Passage: Pitch refers to the frequency of soundwave, and volumn refers to the amplitude of the
soundwave. Besides volumn and pitch, we can also tell the difference between sounds based on the
tone of sound. For example, we can differentiate the sound of different instruments based on the
tone of the sounds.
Claim 1: Volume of sound is the amplitude of the soundwave.
Claim 2: Pitch is the frequency of soundwave.
Claim 3: We can use the tone of the sounds to differentiate the sound of different instruments.

Original question: How are we able to discern whether a sound is coming from in front of us or
behind us?
Passage: There are multiple explanations for why we can localize sounds. One explanation is that
sounds travelling to the corresponding side of one's ear will be slightly louder. Another
explanation is that there is a slight difference in the hitting time to one's left and right ear
based on the sound's direction. However, these explanation means that when a sound is exactly in
front of someone or exactly behind someone, he or she can not tell the difference.
Claim 1: We can localize sounds by recognizing that the sound travelling to the corresponding side
of one's ear will be slightly louder.
Claim 2: We can also localize sounds by recognizing the difference in hitting time to one's left and
right ear based on the sound's direction.
Claim 3: We cannot tell the difference between a sound that is exactly in front of us or exactly
behind us.

Table 22: Prompt used to generate the sub-claims for ELI5 questions. Blue text is model generation. Brown text
is the ELI5 example that we want to generate sub-claims for. We construct the prompt by manually writing the
sub-claims for three questions from the training set.

Instruction: Write an accurate, engaging, and concise answer for the given question using only
the provided search results (some of which might be irrelevant) and cite them properly. Use an
unbiased and journalistic tone. Always cite for any factual claim. When citing several search
results, use [1][2][3]. Cite at least one document and at most three documents in each sentence.
If multiple documents support the sentence, only cite a minimum sufficient subset of the
documents.

Table 23: Instruction for VANILLA.

Instruction: Write a high-quality answer for the given question using only the provided search
results and cite them properly using [1][2][3].

Table 24: Short instruction for VANILLA.

---

Summarize the following document within 50 words with the question of interest "{QUESTION}"
Return "irrelevant" if the document is irrelevant to the question. Try to keep all the important
dates, numbers, and names.

Title: {TITLE}
Text: {TEXT}
Summary:

---

Table 25: Prompts for SUMM.

---

Given the follow passage and the question "{QUESTION}", extract a useful span from the passage
that can answer the question. Resolve all the coreference issues to make the extracted span
understandable standalone. If the passage is not helpful for answering the question, return
"irrelevant".

Title: {TITLE}
Text: {TEXT}
Extracted span:

---

Table 26: Prompts for SNIPPET.

---

Instruction: Write an accurate, engaging, and concise answer for the given question using only
the provided search results and cite them properly. Use an unbiased and journalistic tone.
Always cite for any factual claim.
You are provided summaries/snippets of the search results. You can use "Check: Document [1][2]"
to check the corresponding full documents (you should only check relevant documents and you can
at most check 3 documents at a time) and use "Output:" to output a sentence in the answer. In the
answer, cite properly by using [1][2][3]. Cite at least one document and at most three documents
in each sentence. If multiple documents support the sentence, only cite a minimum sufficient
subset of the documents. Use "End" to end the generation.

<Retrieve for question "...">
<Get summaries/snippets for the passages and delete those that are "irrelevant">
Document [1](Title: ...) {SUMMARY OR SNIPPET}
...

Question: When did US break away from England?
Check: Document [1][2]
Document [1] {FULL TEXT}
Document [2] {FULL TEXT}
Output: The United States ... **[1]** ... **[2]**.
<Remove the full text of [1][2] from context>
Check: Document [3]
Document [3] {FULL TEXT}
Output: The Treaty of Paris ... **[3]**.
<Remove the full text of [3] from context>
End.

---

Table 27: An example for INTERACT.

---

Instruction: Write an accurate, engaging, and concise answer for the given question using only
the provided search results and cite them properly. Use an unbiased and journalistic tone.
Always cite for any factual claim.
You can use "Search: key words" to check the most relevant document's full text and use
"Output:" to output a sentence in the answer. In the answer, cite properly by using [1][2][3].
Cite at least one document and at most three documents in each sentence. If multiple documents
support the sentence, only cite a minimum sufficient subset of the documents. Use "End" to end
the generation.

---

Table 28: Instruction for INLINESEARCH.

---

Instruction: Write an accurate, engaging, and concise answer for the given question. Use an
unbiased and journalistic tone.

---

Table 29: Instruction for CLOSEDBOOK.

Instruction: Write an accurate, engaging, and concise answer for ...

Document [1](Title: How to Treat and Prevent Food Poisoning - MsPrepper) just a typical gastro upset. Salmonella is most commonly caused by eating undercooked or raw foods like eggs or meat. You know how your mom always warned you not to eat raw cookie dough? This is why. Most people do eat cookie dough and they are fine, but salmonella is a risk. If you do contract salmonella, you could start to feel bad within in a couple of hours after eating contaminated food, and sometimes it could take a day or two. Common symptoms are nausea and vomiting, loose stools (sometimes bloody), flu like symptoms, and stomach cramps. To treat
Document [2](Title: FDA Issues Warning About Eating Raw Cookie Dough, But Not For Salmonella Risks) FDA Issues Warning About Eating Raw Cookie Dough, But Not For Salmonella Risks Used to licking the spoon or placating yourself with full-on chunks of raw cookie dough? The Food and Drug Administration issued a warning on Tuesday that strongly advises against continuing the habit. The agency asserted that consuming raw batter of any kind, whether for bread, cookies or pizza, could make a person sick. While you may have been warned in the past against eating raw dough due to the risk of contracting salmonella from raw eggs, the FDA is citing raw flour as the culprit for a
Document [3](Title: It's Probably OK to Eat Raw Cookie Dough — As Long As You're Smart About It - The Crux - Very Top Secret Information) First, when most people think about health risks and cookie dough, they think about raw egg. Eggs can be contaminated with salmonella bacteria, and food safety recommendations encourage people to cook eggs until the white and yolk are firm in order to kill any bacteria. However, anyone making cookies can do things to reduce this risk by using pasteurized egg products. When my kids and I make cookie dough, we never use regular eggs. Instead, we use shell eggs that have been pasteurized to kill any harmful bacteria without actually cooking the egg itself. (A great public health innovation, if
Document [4](Title: How Dangerous Is It to Eat Raw Cookie Dough? | Men's Health) Can Eating Raw Cookie Dough Really Make You Sick? Scientists reveal the truth about this supposedly dangerous delicacy By Katherine Dempsey There are few things more tempting in life than eyeing a bowl of cookie dough and deciding whether or not to stick your finger in for a scoop. It's a bit like playing Russian roulette. You could get lucky and enjoy the delicious dough without conseqence, but there's always the risk of getting serously sick with a food-borne illness. That's because multiple ingredients within the dough could be contaminated by pathogens such as Salmonella and E. coli, says Soohyoun
Document [5](Title: How Dangerous Is It to Eat Raw Cookie Dough? | Men's Health) is usually pasteurized, so it's not likely that the egg would make you sick. However, other ingredients in the dough could potentially harbor pathogens. Experts say that a prime suspect in a 2009 E. coli outbreak linked to prepackaged cookie dough was actually the flour. Nuts and chocolate have also been linked to Salmonella outbreaks. Bottom line: You're better off skipping raw cookie dough. At least you'll have killer cookies as a consolation. The article How Bad Is It To Eat Raw Cookie Dough? originally ran on Prevention.com The 9 Foods Most Likely to Make You Sick Mmm, Just Don't ...

Question: Why is it bad to eat cookie dough for risk of salmonella but things like Cookie Dough Bites are ok?
Answer: Raw cookie dough is not recommended to be eaten due to the risk of salmonella [1][2]. Eating raw flour is also a risk for food poisoning [2]. The egg in cookie dough is usually pasteurized, so it's not likely that the egg would make you sick, but other ingredients in the dough could potentially harbor pathogens [4][5]. However, prepackaged cookie dough like Cookie Dough Bites is safe to eat because the dough is made with pasteurized egg products and heat-treated flour [2][3]..

Ground truth claims:
1. Cookie Dough Bites are safe to eat since they do not contain eggs.
2. Salmonella is a bacteria found in eggs.
3. Eating raw cookie dough carries the risk of salmonella infection.

Table 30: ELI5 example 1.

Instruction: Write an accurate, engaging, and concise answer for ...

Document [1](Title: Is Snapchat really worth $19 billion? - CSMonitor.com) reporting that the Los Angeles-based company is aiming to raise $500 million at a valuation of $16 billion to $19 billion, making it the third most highly valued tech start-up backed by venture capitalists. The Chinese handset maker Xiaomi is valued at $45 billion, while Uber is estimated to be valued at about $40 billion, according to data from CB Insights. Read MoreVC investment hits $86B thanks to Uber, Xiaomi Snapchat was valued at $10 billion in August, according to a Dow Jones report. Some of its investors from previous rounds include Benchmark, Lightspeed Venture Partners and Kleiner Perkins Caufield Document [2](Title: What Are Venture Capital Investments? – DollarsAndSense.my) Ever wondered how highly valued technology giants like Google and Facebook were able to grow so fast and pay their employees so well in such a short amount of time, or how still growing start-ups like Uber are able to lose 1.2 billion US dollars in just the first half of this year alone and still command a valuation upwards of 50 billion US dollars? The answer lies with a special category of investment activity known as venture capital. Venture capitalists are professional investors who invest in a number of highly scalable high-risk technology ventures hoping to make a multi-fold Document [3](Title: Opinion | What Dara Khosrowshahi Must Do to Save Uber - The New York Times) at a discount. These are troubling signs. Every start-up must one day fulfill the market's demand that it turn a profit, but Uber has never figured out how to do that. While ride sharing in some form will probably survive, it's more likely that without some drastic changes, Uber won't be around in three to five years. Mr. Khosrowshahi must avoid the mistakes of his predecessor by accepting that "pivots" (Silicon Valley-speak for the desperate changes troubled companies make to reassure their venture capitalist funders) are not the answer. None of the pivots Mr. Kalanick tried — like on-demand delivery Document [4](Title: Snapchat raising funding round at $19 billion valuation: Report) Snapchat raising funding round at $19 billion valuation: Report The mobile messaging app Snapchat is said to be seeking funding at a $19 billion valuation, according to a report, citing a person with knowledge of the matter. Bloomberg is reporting that the Los Angeles-based company is aiming to raise $500 million at a valuation of $16 billion to $19 billion, making it the third most highly valued tech start-up backed by venture capitalists. The Chinese handset maker Xiaomi is valued at $45 billion, while Uber is estimated to be valued at about $40 billion, according to data from CB Insights. Document [5](Title: Unicorns And Wall Street | MoneyTips) not start showing profits soon. Even Uber, with its massive $62-billion-plus valuation, will test investor's patience if it does not start showing signs of a profit in the next few years. Cracks are starting to show in terms of pre-IPO devaluations. Dropbox and Snapchat have seen a nearly one-quarter drop in their pre-IPO valuations, while Square's pre-IPO value has dropped by approximately one-third. Larger investors and venture capitalists are sending a message to the unicorns: Outline a solid path to profits and start producing them soon, or expect your valuation to be slashed. Diving ahead with an IPO is not ...

Question: How are firms like snapchat, uber etc valued so highly while still not making a profit? Do venture capitalists not expect some form of repayment within a number of years?
Answer: Venture capitalists invest in a number of highly scalable high-risk technology ventures hoping to make a multi-fold return on their investment in a short amount of time [2]. Highly valued companies like Snapchat, Uber, and Xiaomi are able to grow so fast and command high valuations because they have the potential to generate significant revenue in the future [2][4]. Venture capitalists expect a return on their investment within a number of years, but they also invest in the long-term potential of the company [2]. It is important to note that every start-up must eventually turn a profit, but some start-ups like Uber have not yet figured out how to do that [3][5].

Ground truth claims:
1. Firms like Snapchat and Uber need to establish their brand and amass users before introducing ads.
2. Introducing ads too early can deter potential users.
3. Uber is reinvesting a lot of money to make their service better.

Table 31: ELI5 example 2.