# OpenReview forum: "Enabling Large Language Models to Generate Text with Citations"
_EMNLP/2023/Conference — EMNLP 2023 Main_

### Official Review · Reviewer_jGRf · 2023-07-21

**Soundness:** 4

**Excitement:**

4: Strong: This paper deepens the understanding of some phenomenon or lowers the barriers to an existing research direction.

**Paper Topic And Main Contributions:**


the authors present a method for generating citations with LLMs as a means to battle factual correctness and verifiability. They provide a benchmark for citation evaluation and use automatic metrics for fluency, correctness and citation quality. These metrics correlate well with human judgements. Nice contribution to the field.

**Questions For The Authors:**

- exact match seems very strict, what if, for example a date was formatted in a different way from the exact answer, or are there multiple answer models per question?
- recall 100% if at least 5 correct answers is given - but I assume also 100% if there are less than 5 relevant and all are given?
- Post-editing; why n=4 answers?
- in terms of retrieval analysis. Is there any effect in the overlap of selected passages on correctness? Aka if the retrieved passages overlap a lot is the correctness then higher?
- Would it be possible to instruct chatGPT to reuse the text from the passages in order to maybe improve the exact match?

**Reasons To Accept:**

- important topic, very relevant for current challenges
- well written, thorough work

**Reasons To Reject:**

unclear wether the dataset (at least the randomly selected examples) will be released - without it, it would be hardly a reproducible benchmark. (for the remainder I have assumed that they will release it as they do state "We present ALCE, the first reproducible benchmark" and the rest of the article is very nice)

minor point:
a little bit unclear how the kappa is calculated between ALCE (binary?) and human judgement (3 classes) for citation precision

**Reproducibility:**

4: Could mostly reproduce the results, but there may be some variation because of sample variance or minor variations in their interpretation of the protocol or method.

**Reviewer Confidence:**

4: Quite sure. I tried to check the important points carefully. It's unlikely, though conceivable, that I missed something that should affect my ratings.

---

> ### Author Rebuttal · Authors · 2023-08-28
>
> Thanks for your feedback! We are encouraged that you found ALCE important and relevant, and our work thorough and well written. We answer your questions below.
>
> **(1) Unclear whether the dataset will be released.**
>
> We attached our data as well as evaluation code in the supplementary material. We will release them upon paper acceptance.
>
> **(2) Unclear how the kappa is calculated between ALCE (binary?) and human judgement (3 classes) for citation precision.**
>
> Sorry for the confusion. As stated in Section 6 (line 476-478), in human evaluation, we consider a citation to have a precision score of 1 if the citation “partially supports” or “fully supports” the statement. Then, we take the ALCE citation precision and human citation precision results (both binary) for calculating the Kappa.
>
> **(3) Exact match seems very strict. What if, for example, a date was formatted in a different way from the exact answer, or are there multiple answers per question?**
>
> Both ASQA and QAMPARI provide aliases for each entity answer. We normalize the model outputs and answers (deleting all punctuations and lower-casing all letters) and report the exact-match with the best matching aliases (line 962). Most ASQA and QAMPARI short answers are entity names or dates so exact match with aliases should suffice.
>
> **(4) Recall is 100% if at least 5 correct answers are given. Is the recall also 100% if there are fewer than 5 correct answers and all are given?**
>
> Your understanding is correct. If all correct answers are given, the recall is also 100%.
>
> **(5) Why do you choose n=4 for Rerank?**
>
> In our preliminary experiments, we find that n=4 achieves a good balance between citation quality, correctness, and inference speed. Using larger n leads to marginal improvement on citation quality but is significantly slower.
>
> **(6) In terms of the retrieval analysis, is there any effect in the overlap of selected passages on correctness? Aka if the retrieved passages overlap a lot is the correctness then higher?**
>
> This is a great question. We did not conduct such an analysis but will make sure to add it in the revision. Our intuition is that a more diverse set of retrieved passages will benefit the correctness, as most questions require multiple different sources to answer. This is also corroborated by findings in ASQA (Stelmakh et al., 2022).
>
> **(7) Would it be possible to instruct ChatGPT to reuse the text from the passages in order to maybe improve the exact match?**
>
> This is a good point. In our preliminary experiments, we tried a similar prompting strategy where we asked ChatGPT to explicitly copy the text from the retrieved passages, but it did not lead to an improvement in correctness.
>
> Stelmakh et al. ASQA: Factoid Questions Meet Long-Form Answers. EMNLP 2022.

---

### Official Review · Reviewer_D8zW · 2023-08-02

**Soundness:** 4

**Excitement:**

4: Strong: This paper deepens the understanding of some phenomenon or lowers the barriers to an existing research direction.

**Paper Topic And Main Contributions:**

In this paper, the authors present a novel approach to LLMs where citations are required for specific text passages that LLM generates.

The introduction of this feature (citation) offers advantages such as verify the claims made by LLMs (by referring to the provided citations), and reduce hallucination.

The key results of this study are that: the quality of retrieval plays a pivotal role in determining the final performance, the limited context window of LLMs restricts the number of passages they can integrate, LLMs encounter difficulties in effectively synthesizing multiple documents in context while avoiding distraction from irrelevant information.


**Questions For The Authors:**

Could you provide the code used in this experiments and share the assessments performed by the annotators for reproducibility purposes?

**Reasons To Accept:**

Besides the core topic addressed by the paper (adding citations to the generated text, which is one of the most important problems of LLMs), the experimental setup and analysis are very well presented.

The quality of the output which is produced by LLMs is analyzed according to three different dimensions (fluency, correctness, quality of citations).

The human evaluation is an added value to this paper in order to double check the quality of the results.

**Reasons To Reject:**

There are a couple of minor issues that I would like to point out:
- why do you "We randomly select 1,000 examples from the development set of each dataset for ALCE."?
- how do you measure citation recall?  I mean, recall in terms of total number of citations that support the claim. It is not clear even from the papers that are cited in Section 3.3.



**Reproducibility:**

3: Could reproduce the results with some difficulty. The settings of parameters are underspecified or subjectively determined; the training/evaluation data are not widely available.

**Reviewer Confidence:**

3: Pretty sure, but there's a chance I missed something. Although I have a good feel for this area in general, I did not carefully check the paper's details, e.g., the math, experimental design, or novelty.

---

> ### Author Rebuttal · Authors · 2023-08-28
>
> Thanks for your feedback! We are encouraged that you found our experiments and analysis thorough and that the human evaluation helped validate our metrics. We answer your questions below.
>
> **(1) Why do you randomly select 1,000 examples from the development set of each dataset for ALCE?**
>
> We randomly sample examples mainly for efficiency. Running inference of LLMs can be slow and costly; furthermore, in our preliminary experiments, using more examples for evaluation does not change the results or the variance significantly.
>
> **(2) How do you measure citation recall?**
>
> Sorry for any confusion in the paper. Citation recall is defined as “the percentage of output statements that are fully supported by their citations”. We use TRUE (Honovich et al., 2022) to determine whether the cited passages support the statement. TRUE is a T5 model fine-tuned on multiple natural language inference (NLI) datasets.
>
> An NLI model takes a “premise” and a “hypothesis” and outputs whether the “premise” entails the “hypothesis”. We concatenate all cited passages as the “premise”, and take the statement as the “hypothesis”. We say the passages support the statement if the NLI model outputs “entailment”. Our Figure 3 provides an example for the process.
>
> **(3) Could you provide the code used in this experiments and share the assessments performed by the annotators for reproducibility purposes?**
>
> We provided our code, data, and evaluation instructions in the supplementary materials. We will release them as well as the full annotator assessment logs upon acceptance.
>
> Or Honovich et al. TRUE: Re-evaluating factual consistency evaluation. NAACL 2022.

---

### Official Review · Reviewer_CvfW · 2023-08-06

**Soundness:** 3

**Excitement:**

3: Ambivalent: It has merits (e.g., it reports state-of-the-art results, the idea is nice), but there are key weaknesses (e.g., it describes incremental work), and it can significantly benefit from another round of revision. However, I won't object to accepting it if my co-reviewers champion it.

**Paper Topic And Main Contributions:**

The paper "Enabling Large Language Models to Generate Text with Citations" presents an interesting problem on a proposed benchmark and solutions that leverage LLMs. After the LLM application, the authors apply re-ranking and post-citing.

**Questions For The Authors:**

Question A: Why is there no comparison of the presented results with some existing work in the literature?
Question B: What is the extent of hallucinations, bias, fairness etc. in the text generated by LLMs in the task?

**Reasons To Accept:**

The paper presents an interesting problem with the benchmark -- ALCE, the first benchmark for Automatic LLMs' Citation Evaluation, which has some novelty over previous benchmarks.

The evaluation is done considerating fluency, correctness, and citation quality.

It highlights the efficacy as well as the challenges of LLMs in this task.

Human evaluations add more quality to the benchmark.

**Reasons To Reject:**

Apart from LLMs, no other baseline models are considered for comparison; this makes it difficult the gauge the relative efficacy of the LLMs in light of the state of the art. It is important to know if the LLMs advance the existing literature in performance.

I would have loved more analysis of factors like hallucinations, bias, fairness etc. of LLMs in this task

These have been now addressed in the rebuttal.

**Reproducibility:**

4: Could mostly reproduce the results, but there may be some variation because of sample variance or minor variations in their interpretation of the protocol or method.

**Reviewer Confidence:**

3: Pretty sure, but there's a chance I missed something. Although I have a good feel for this area in general, I did not carefully check the paper's details, e.g., the math, experimental design, or novelty.

---

> ### Author Rebuttal · Authors · 2023-08-28
>
> Thanks for your feedback! We are encouraged that you found our benchmark ALCE novel and useful for highlighting the challenges of LLMs. We address your questions below.
>
> **(1) Why is there no comparison of the LLM results with some existing work in the literature?**
>
> We focus on LLMs because they become popular tools for information-seeking due to their ability to generate engaging output and handle a variety of topics. Without using LLMs, it remains challenging to generate fluent and coherent text and to generalize to a diverse set of questions.
> Our paper tackles a new setting — LLM generations with citations.
>
> Thanks to the reviewer’s suggestion, we added experiments of a state-of-the-art question answering model, Fusion-in-Decoder (FiD; Izacard and Grave, 2021), trained on the training set of ASQA. We train a T5-base FiD model for 5 epochs on the ASQA training set with a batch size of 64 and a learning rate of 1e-4.  Since **there is no citation annotation data available**, we train the model to generate the answer and use our PostCite method (Section 4.3) to generate citations. We evaluate this model on both ASQA (in-domain) and ELI5 (out-of-domain).  **Note that this is not a direct comparison, as ALCE assumes only evaluation data available and uses only few-shot data for prompting**.
>
> ASQA results:
>
> | Model | Fluency (MAUVE) | Correctness (EM recall) | Citation recall | Citation precision
> | -------- | -------- | -------- | -------- | -------- |
> | ChatGPT Vanilla  | 66.6  | 40.4  |  73.6   | 72.5                |
> | FiD + PostCite| 75.8 |   28.4             |  58.1                 | 58.0 |
>
>
>
> ELI5 results:
>
> | Model | Fluency (MAUVE) | Correctness (Claim recall) | Citation recall | Citation precision
> | -------- | -------- | -------- | -------- | -------- |
> | ChatGPT Vanilla  | 57.2  | 12.0  |  51.1  | 50.0                |
> | FiD + PostCite | 25.2 | 4.4          |  39.3                 | 39.3 |
>
> As the results show, the FiD baseline still significantly lags behind prompting ChatGPT in both correctness and citation quality (even though it is trained on 4000+ examples). When tested on another dataset (ELI5), FiD performs even worse, showing that it is challenging to solve the problem by fine-tuning a small pre-trained model.  We are happy to add these results and more discussion in the revision.
>
> We are unsure what the reviewer refers to as “existing literature”. As we stated in the paper, there are no available baselines to compare to for LLM generations with citations, and there is no available training data. WebGPT (Nakano et al., 2021) and GopherCite (Menick et al., 2022) take a similar setting, but their training data, code, and models are not released, and thus it is impossible to reproduce them.
>
> The main contribution of our work is to propose the first automatic benchmark that can evaluate generations with citations, and by evaluating several simple but effective baselines, we demonstrate promising directions to work on. We look forward to new methods emerging to tackle this problem.
>
> **(2) What is the extent of hallucinations, bias, fairness etc. in the text generated by LLMs in the task?**
>
> Our correctness and citation quality evaluations are about hallucination — they examine whether the outputs are factually correct and faithful to the retrieved evidence. We believe that other factors, such as bias and fairness, are beyond the scope of the work.
>
> To provide more error analyses of LLMs’ outputs on our benchmark, we manually examined 30 outputs generated by ChatGPT Vanilla on ASQA and logged different error types here (one output can have multiple types of errors).
>
> | Error type | Occurrences (out of 30 outputs) |
> | -------- | -------- |
> | Missing a short answer because of the retrieved passages do not contain it  | 11 |
> | Missing a short answer while the retrieved passages contain it  | 6 |
> | Hallucination | 2 |
> | Correct but irrelevant content to the question | 2 |
>
> Most of the errors are caused by the low coverage of the retrieval passages. LLMs also fail to extract the useful information from the passages often. There are only 2 instances among all the 30 questions we examined where the LLM made up some factually incorrect contents.
>
> Nakano et al., 2021. WebGPT: Browser-assisted question answering with human feedback.
>
> Menick et al., 2022. Teaching language models to support answers with verified quotes.
>
> Izacard and Grave, 2021. Leveraging Passage Retrieval with Generative Models for Open Domain Question Answering.

---

### Meta-Review · Area_Chair_Rksa · 2023-09-19

**Recommendation:** 4

**Metareview:**

This paper proposed generating citations with LLMs as a means to enhance factual correctness and verifiability. A benchmark for citation evaluation is created which uses automatic metrics for fluency, correctness, and citation quality. Besides the core topic addressed by the paper, the experimental setup and analysis are very well presented. In general, it is a nice contribution to the field.

---

### Decision · Program_Chairs · 2023-10-07

**Decision:**

Accept-Main

**Comment:**

This paper proposed generating citations with LLMs as a means to enhance factual correctness and verifiability. A benchmark for citation evaluation is created which uses automatic metrics for fluency, correctness, and citation quality. Besides the core topic addressed by the paper, the experimental setup and analysis are very well presented. In general, it is a nice contribution to the field.